# The multiplicity of thioredoxin systems meets the specific lifestyles of Clostridia

Cyril Anjou[1], Aurélie Lotoux[1], Anna Zhukova[2], Marie Royer[1¤a], Léo C. Caulat[1], Elena Capuzzo[1¤b], Claire Morvan[1], Isabelle Martin-Verstraete [1,3]*

1 Institut Pasteur, Université Paris Cité, UMR CNRS 6047, Laboratoire Pathogenèse des Bactéries Anaérobies, Paris, France, 2 Institut Pasteur, Université Paris Cité, Bioinformatics and Biostatistics Hub, Paris, France, 3 Institut Universitaire de France, Paris, France

¤a Current address: Institut Pasteur, Université Paris Cité, UMR CNRS 6047, Unité Écologie et Évolution de la Résistance aux Antibiotiques, Paris, France
¤b Current address: Institut Pasteur, Université Paris Cité, UMR CNRS 6047, Unité de Recherche Yersinia, Département de Microbiologie, Paris, France
* isabelle.martin-verstraete@pasteur.fr

**Data Availability Statement:** All data are in the manuscript and/or supporting information files.

**Funding:** This work was supported by the Fondation pour la Recherche Médicale (#ECO202006011710) for the funding of the PhD

## Abstract

Cells are unceasingly confronted by oxidative stresses that oxidize proteins on their cysteines. The thioredoxin (Trx) system, which is a ubiquitous system for thiol and protein repair, is composed of a thioredoxin (TrxA) and a thioredoxin reductase (TrxB). TrxAs reduce disulfide bonds of oxidized proteins and are then usually recycled by a single pleiotropic NAD(P) H-dependent TrxB (NTR). In this work, we first analyzed the composition of Trx systems across Bacteria. Most bacteria have only one NTR, but organisms in some Phyla have several TrxBs. In Firmicutes, multiple TrxBs are observed only in Clostridia, with another peculiarity being the existence of ferredoxin-dependent TrxBs. We used *Clostridioides difficile*, a pathogenic sporulating anaerobic Firmicutes, as a model to investigate the biological relevance of TrxB multiplicity. Three TrxAs and three TrxBs are present in the 630Δ*erm* strain. We showed that two systems are involved in the response to infection-related stresses, allowing the survival of vegetative cells exposed to oxygen, inflammation-related molecules and bile salts. A fourth TrxB copy present in some strains also contributes to the stress-response arsenal. One of the conserved stress-response Trx system was found to be present both in vegetative cells and in the spores and is under a dual transcriptional control by vegetative cell and sporulation sigma factors. This Trx system contributes to spore survival to hypochlorite and ensure proper germination in the presence of oxygen. Finally, we found that the third Trx system contributes to sporulation through the recycling of the glycine-reductase, a Stickland pathway enzyme that allows the consumption of glycine and contributes to sporulation. Altogether, we showed that Trx systems are produced under the control of various regulatory signals and respond to different regulatory networks. The multiplicity of Trx systems and the diversity of TrxBs most likely meet specific needs of Clostridia in adaptation to strong stress exposure, sporulation and Stickland pathways.

of CA, by the Institut Universitaire de France for IMV and by the ANR Difox (ANR-22-CE15-0026-01) for the salary of AL. We received also financial support from Institut Pasteur and Université Paris Cité. The funders had no role in study design, data collection and analysis, decision to publish, or preparation of the manuscript.

**Competing interests:** The authors have declared that no competing interests exist.

## Author summary

Cells, that are exposed to oxidative stress in their environment, must rapidly adapt and repair their proteins and thiols. The thioredoxin (Trx) system plays a crucial role in the protection of cysteine from oxidation. Despite being ubiquitous, their role in the obligate anaerobic Clostridia, the relevance of the atypical multiplicity of Trx reductases in bacterial physiology and the importance of a clostridial-specific ferredoxin-dependent Trx reductase had remained unexplored. We analyzed the role of the thiol repair Trx systems in the gut enteropathogen *Clostridioides difficile*, a major cause of antibiotic-associated diarrhea. Two Trx systems are involved in the response to stresses encountered in the gastrointestinal tract during infection. One of these Trx systems is also present in the spore, the form of transmission and persistence in the environment, and protects the spore from hypochlorite, a disinfectant used to eradicate the spores in hospital. The third system is involved in glycine catabolism and contributes to efficient sporulation. This multiplicity of enzymes seems to meet the needs of cell during growth, compartmentation, and differentiation, not only in Clostridia but perhaps in other multiple-Trx reductase organisms such as Cyanobacteria or eukaryotes, which have dedicated Trx systems in mitochondria and chloroplast.

## Introduction

Most cells are exposed to oxidative stress in their environment [1], either to oxygen ($O_2$) itself or to exogenous and endogenous sources of reactive oxygen species (ROS). Microorganisms must adapt to these oxidative conditions that lead to cellular damages with the oxidation of proteins, membrane components and nucleic acids [2–4]. Protein oxidation occurs mainly on cysteine residues, causing the formation of disulfide bonds, of sulfenic (R-SOH) or sulfinic (R-SO$_2$H) acids but also the irreversible formation of sulfonic acid (R-SO$_3$H). In addition, the presence of nitric oxide (NO) or reactive nitrogen species can cause S-nitrosylation of cysteine [2,5,6]. These modifications can inactivate proteins, and cells thus require active repair strategies that include thioredoxin (Trx) systems. Trx systems are ubiquitous thiol- and protein-repair systems, that play a crucial role in oxidative and nitrosative stress resistance [2,5,7]. These systems are composed of a thioredoxin (TrxA) and a thioredoxin reductase (TrxB) [5]. TrxAs are small proteins with a CXXC redox active site allowing disulfide bond exchange reactions for repair of oxidized cysteines in proteins *via* the reduction of intra- or inter-molecular disulfide bonds, sulfenic acid or S-nitrosylation in target proteins [5,8]. The resulting oxidized TrxAs are then recycled by a TrxB, usually a FAD-dependent protein in prokaryotes, that uses a reduced substrate, commonly NADPH (less frequently NADH) [5]. In model bacteria where Trx systems have already been studied, a unique and versatile NAD(P)H-using Thioredoxin Reductase (NTR) reduces a variable number of TrxAs. One to six TrxA proteins are present in a single microorganism [9]. However, atypical Trx systems that differ from canonical ones by their co-factors (FAD or Fe-S clusters), their electron donors or their organizations with fused TrxA and TrxB proteins exist [10]. TrxBs using a ferredoxin instead of NAD(P)H as electron donor have been described in Clostridia [5,11]. These organisms harboring Ferredoxin-dependent Flavin Thioredoxin Reductase (FFTRs) could have several TrxBs instead of a unique pleiotropic reductase [11]. The role of Trx systems in Clostridia, the relevance of the multiplicity of TrxBs in bacterial physiology and the role and importance of FFTRs remain unexplored so far.

Clostridia is a diverse polyphyletic group that includes ubiquitous bacteria of environmental, medical and biotechnological interest including gut commensals and pathogens [12–14]. One of these gut pathogens is *Clostridioides difficile*, which exhibits characteristics shared by Clostridia—Gram-positive, anaerobic, low GC content and usually spore-formers—as well as metabolic pathways such as Stickland reactions present in proteolytic Clostridia [15,16]. *C. difficile* causes moderate to severe diarrheas, pseudomembranous colitis or toxic megacolon, which can lead to the death of the patient [17,18]. *C. difficile* infection (CDI) classically occurs following antibiotic-induced gut dysbiosis [19]. The altered composition of the gut microbiota results in substantial changes in metabolic pools, with a decrease of secondary bile acids due to the loss of primary bile acids-metabolizing commensal bacteria [20,21]. While secondary bile acids are toxic for *C. difficile*, preventing host colonization by this organism [22], most of primary bile acids allow spore germination [23], which is followed by vegetative cell multiplication. Concomitantly, the dysbiosis leads to a depletion of butyrate-producing bacteria, inducing a metabolic switch in colonocytes. This switch, from the oxidation of butyrate towards glucose fermentation that consumes less $O_2$, increases the $O_2$-tension in the gut [24] contributing to a decrease of the integrity of the intestinal barrier [25]. In addition, the depletion of secondary bile acids also promotes inflammation [26]. These effects are amplified during CDI as a result of the action of the *C. difficile* toxins [27]. Their activity causes a disruption of the epithelial intestinal barrier [28–30], triggering a strong immune response with the secretion of pro-inflammatory cytokines and chemokines, the recruitment of immune cells and the production of various antimicrobial compounds notably ROS and reactive nitrogen species [28,31,32]. The activity of the toxins and the resulting host-response lead to the symptoms of the infection [31]. *C. difficile* survival to this inflammatory reaction of the host is supported by a large arsenal of detoxication enzymes [33,34] and repair systems that include Trx systems. Trx systems seem to be the only ones ensuring protein repair as *C. difficile*, like most Clostridia, lacks synthesis pathways and reductases associated with other known thiol repair systems, *e.g.*, glutathione, mycothiol and bacillithiol [35–37]. Even if an uncharacterized thiol repair system may exist, the absence of these conventional alternative systems suggests a crucial role of Trx systems in *C. difficile* physiology.

Here, we studied the phylogenetic distribution of Trx systems in prokaryotes and the biological relevance of harboring multiple Trx systems. Specifically, we show the dedication of Trx systems in the key steps of the life cycle of *C. difficile* and their associated regulations. This study provides new insights into the physiology of this anaerobic gut pathogen and sheds light into the atypical complexity of Clostridial Trx systems.

## Results

### Complexity of Clostridial Trx systems

We analyzed the composition of Trx systems in a database containing 349 genomes representative of the bacterial diversity from Megrian *et al.* [38] using BLAST [39] with TrxA and TrxB sequences from *Escherichia coli* and *C. difficile*. Distinction between NTR and non-NTR was made by identification of the NAD(P)-binding site [11] (S1A Fig) in TrxB sequences. Results were then reported on a schematic tree of the prokaryote phylogeny [40] (Fig 1A). A high variability of the number of TrxAs is observed as previously described [9,41,42]. Conversely, most of the bacteria, including those for which Trx systems have been extensively studied such as *E. coli*, *Mycobacterium tuberculosis* and *Helicobacter pylori*, harbor a unique TrxB. Some phyla contain more than one TrxB, which is often correlated with the presence of a non-NTR. Among them, Firmicutes is the most remarkable one with up to four TrxBs in the same bacterium. Within this phylum, a diversity of the number of TrxBs is observed, with only one TrxB for the Bacilli and Tissierellales, and two to four copies for the Negativicutes and Clostridia

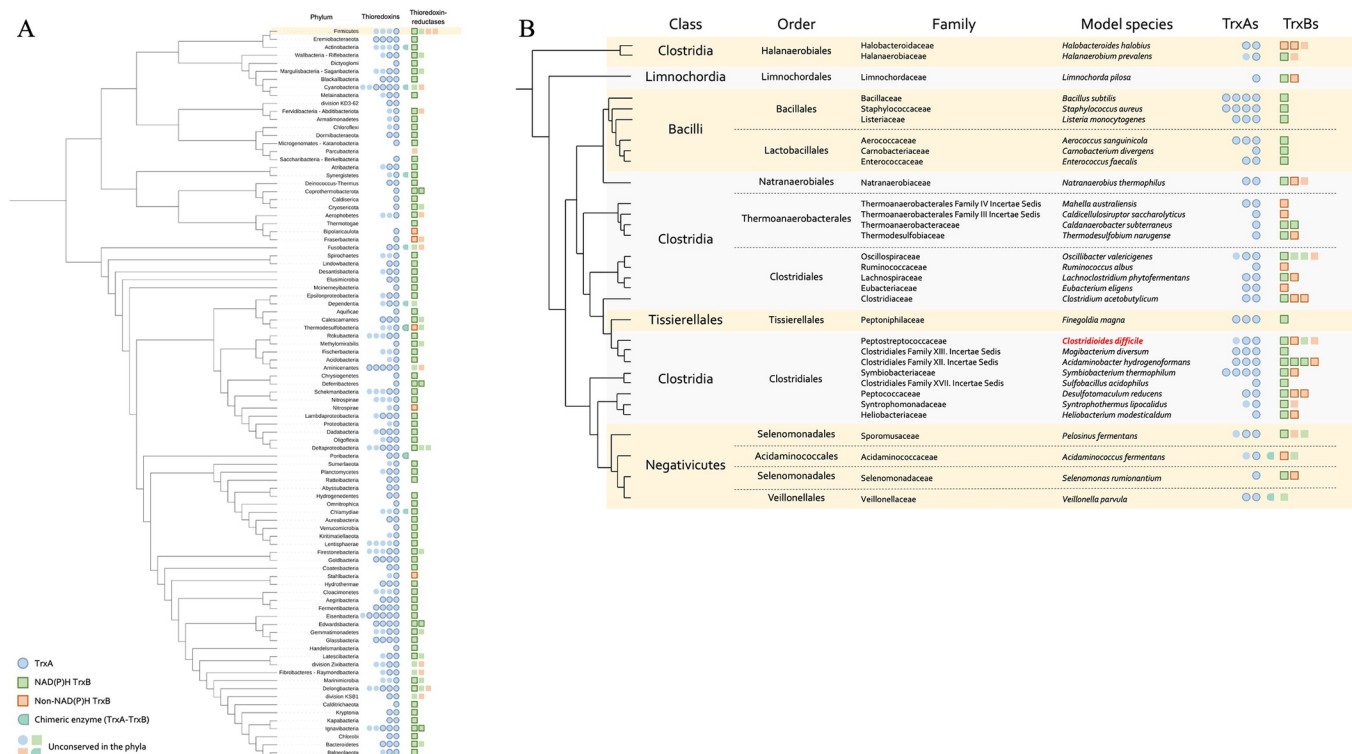

**Fig 1. Composition of Trx systems across Bacteria.** Composition of the Trx systems among the bacterial phylogeny (A) or among Firmicutes (B). The schematic phylogeny is based on Witwinosky *et al.* [40] (A) and on Taib *et al.* [43] (B). TrxA and TrxB are indicated by circles and squares, respectively. NTR are indicated in green and non-NTR in orange. Dark border indicates conservation among the phyla (A) or within the family (B), forms without border indicate presence in some individuals of the phyla but not in all of them. Chimeric enzymes are proteins with fused TrxA and TrxB domains. Firmicutes are highlighted in yellow (A). *C. difficile* is highlighted in bold red.

with a few exceptions (Fig 1B). In the latter two groups, most of the bacteria have both types of TrxB: NTR and non-NTR, which are probably FFTR [11]. Despite the numerous studies on the role of Trx systems in bacterial physiology, their role in Clostridia and the implication of the presence of multiple systems have never been studied.

We focused on *C. difficile*, a genetically manipulatable bacterium with a well-characterized lifecycle. The reference strain (630Δ*erm*) of our model bacterium has three TrxAs (TrxA1/CD630_16900, TrxA2/CD630_30330, TrxA3/CD630_23550) and three TrxBs (TrxB1/CD630_16910, TrxB2/CD630_21170, TrxB3/CD630_23560). TrxB2 and TrxB3 harbor NAD(P)H binding sites with GxGxxA and HRRxxxR motifs [11], suggesting that these two proteins are NTRs (S1A Fig). By contrast, TrxB1 lacks these NAD(P)H binding motifs and clusters with other clostridial FFTRs, suggesting that this copy uses ferredoxin as cofactor (S1A–S1B Fig). Finally, TrxA1 and TrxA2 harbor a classical TrxA WCGPC motif [5], while TrxA3 presents an atypical GCEPC motif described in other Clostridia [44] (S1C Fig).

## Regulation of *C. difficile* Trx systems

A first step in deciphering the complexity of Trx systems is the analysis of their genetic organization and the regulation of their expression. The *trxA1* and *trxB1* genes form a two-gene operon, while *trxA3* and *trxB3* are part of the *grd* operon encoding the glycine-reductase [45] (Figs 2A and S1D). By contrast, the *trxA2* and *trxB2* genes are monocistronic and located in two distinct loci on the chromosome. Analysis of the genome-wide transcription-start site

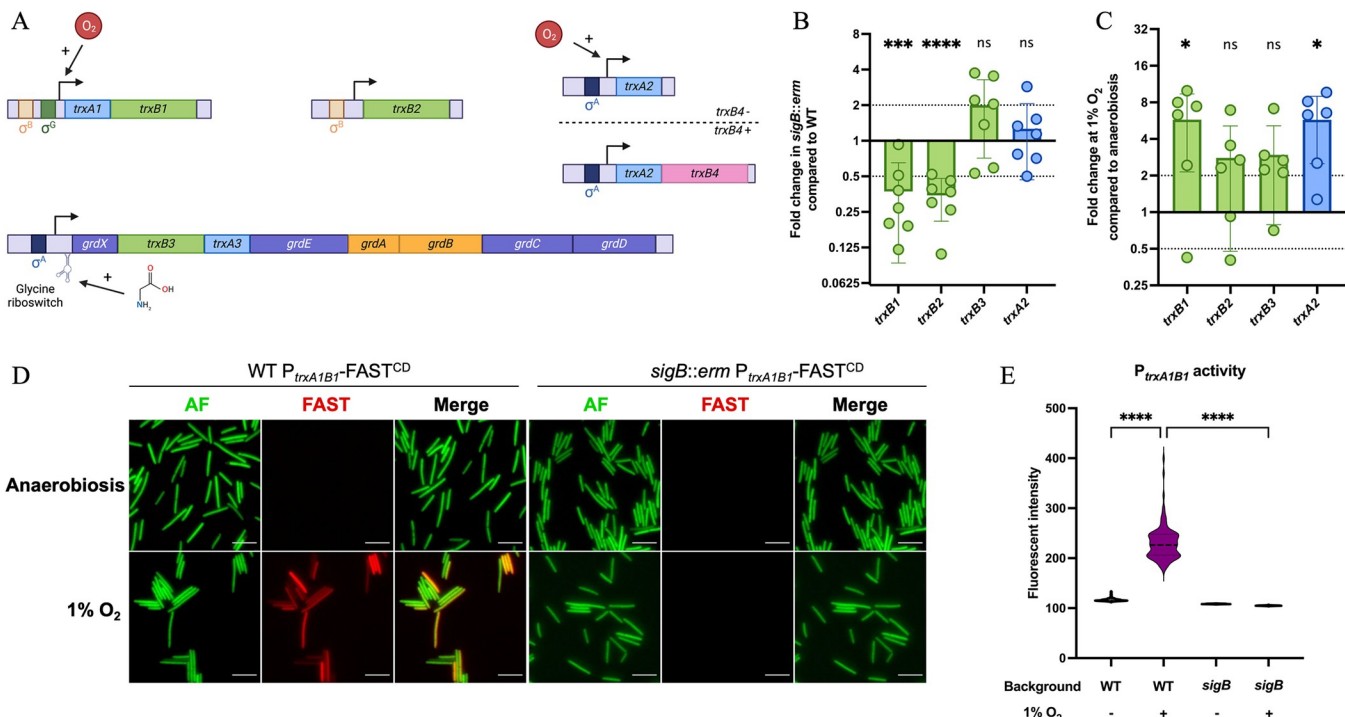

**Fig 2. Expression of *C. difficile* Trx systems.** (A) Genetic organization and transcriptional regulation of *trx* genes. The four *trx* loci are represented, with *trxA* genes in light blue and *trxB* genes in green. The two organizations of *trxA2* (in *trxB4* positive and negative strains) are represented with the additional *trxB4* represented in pink. (B, C) Expression of *trx* genes was measured by qRT-PCR in (B) WT strain and *sigB* mutant after 4.5 h of growth in TY and in (C) WT strain after 24 h of growth in TY in anaerobiosis or at 1% $O_2$. Mean and Standard Deviation (SD) are shown. Experiments were performed in 6 biological replicates. One sample t-tests were used with comparison of the fold change to 1. (D) Pictures of *C. difficile* WT and *sigB* mutant strains carrying a transcriptional fusion $P_{trxA1/B1}$-FAST$^{CD}$. Cells were cultivated during 24 h in TY liquid medium in anaerobic or hypoxic conditions (1% $O_2$) and observed at high magnification (60X). Images are overlays of bacterial autofluorescence (AF) (green) and FAST (red). Scale bars represent 10 μm. (E) Average $P_{trxA1B1}$-FAST$^{CD}$ fluorescence intensity of individualized bacteria from acquired images of panel D. Each group consists in the measure of the average $P_{trxA1B1}$-FAST$^{CD}$ of 600 cells from two independent experiments. Kruskal-Wallis tests were performed followed by Dunn's multiple comparison test. *: p-value<0.05, ** <0.01, *** <0.001 and **** <0.0001.

(TSS) mapping data [46] shows the presence of $\sigma^A$-recognized promoters upstream of the TSSs of *trxA2* and the *grd* operon containing *trxA3B3*. 5'RACE technique was performed to identify the TSSs of the *trxA1B1* operon and the *trxB2* gene. Both TSSs were preceded by consensus sequences corresponding to promoters recognized by $\sigma^B$, the sigma factor of the general stress-response [33] (S2A Fig). We then compared by qRT-PCR the expression of the *trx* genes in the wild-type (WT) 630Δ*erm* strain and in the *sigB*::*erm* mutant [33] (Fig 2B). As expected, no significative differences were observed for the expression of the *trxB3* and *trxA2* genes in the two strains, while the expression of the *trxB1* and *trxB2* genes decreased in the *sigB* mutant compared to the WT strain, confirming the $\sigma^B$-dependent control of these genes.

As *C. difficile* faces low $O_2$ tensions in the gut [47], and because of the role of Trx systems in oxidative stress response in other bacteria, we investigated the effect of $O_2$ on the expression of *trx* genes. Using qRT-PCR, we showed that exposure to 1% $O_2$ for 24 h led to a significant increase of expression of the *trxB1* and *trxA2* genes while the expression of *trxB2* and *trxB3* remained unchanged (Fig 2C). To confirm the induction of *trxA1B1* expression, we used a transcriptional fusion between the promoter of this operon and the FAST$^{CD}$ reporter gene [48,49]. The FAST system is $O_2$ independent and functions both in anaerobiosis and in the presence of $O_2$. This system has been successfully used in *Clostridium acetobutylicum* [49]. While expression of the $P_{trxA1B1}$-FAST$^{CD}$ fusion was not detected in anaerobiosis, it was

significatively induced in the presence of 1% of $O_2$ (Fig 2D and 2E). The absence of fluorescent signal in the presence of $O_2$ in the *sigB*::*erm* mutant confirmed the $\sigma^B$-dependent control of this operon. Finally, we detected in the WT strain a heterogeneity of expression of the $P_{trxA1B1}$-FAST$^{CD}$ fusion, as observed previously for other $\sigma^B$ targets [50].

In summary, the *trx* loci are differentially regulated (Fig 2A). The *trxA1B1* operon is $\sigma^B$-regulated and is induced in presence of $O_2$, while *trxB2* is not induced by $O_2$ despite its $\sigma^B$-dependent regulation. The *trxA2* gene is also induced in presence of $O_2$ but is transcribed from a $\sigma^A$-dependent promoter, indicating that $O_2$ induction and $\sigma^B$-dependent control are not linked. Finally, the *trxA3B3* genes are part of the *grd* operon and are likely transcribed by $\sigma^A$. A glycine riboswitch has been previously identified upstream of the *grd* operon [51], suggesting an induction by glycine, as demonstrated in a recent study [52].

## Trx systems are not essential in *C. difficile*

To investigate the role of the three Trx systems in *C. difficile* physiology, we inactivated the different *trx* genes. We were able to obtain mutants for all *trx* genes except *trxB3* despite several attempts, and all the corresponding multi-mutants were obtained. We also obtained complemented strains for the different mutants with plasmids carrying the corresponding genes under the control of their own promoters. We were able to obtain a triple *trxA* mutant indicating that Trx systems are not essential in *C. difficile*. In addition, no growth defects were observed for any of the single or multiple mutants in TY medium (S3A Fig). However, the survival of the *trxA* triple mutant in stationary phase was strongly impacted, with a defect of more than two-logs already visible at 24 h (S3B Fig). The survival of the $\Delta trxA1$/$\Delta trxA2$ double mutant was also affected, but to a lesser extent than the triple mutant. No survival defect was observed for the *trxB* mutants (S3C Fig). These results indicated that even if TrxAs are not necessary for *C. difficile* growth, they are important for its survival during stationary phase, with a more important role of TrxA1 and TrxA2 compared to TrxA3.

## Two Trx systems favor the survival of *C. difficile* when exposed to $O_2$ and inflammation-related stresses

Even though *C. difficile* is a strict anaerobe and a gut resident, this pathogen is exposed to various low $O_2$ tensions during its infectious cycle [47]. *C. difficile* must also deal with immune cells [32] and inflammation-produced molecules [27]. All these stresses target principally thiols [53], and we demonstrated that some *trx* genes were induced by low $O_2$ tensions or controlled by $\sigma^B$. Thus, we investigated the role of Trx systems in stress tolerance by assessing the survival of all the *trx* mutants upon exposure to different molecules.

We first evaluated the survival in the presence of 1% $O_2$ to mimic the $O_2$-level at the mucus layer [47] and at 0.1% of $O_2$ which corresponds to $O_2$-tension in a dysbiotic gut lumen [47]. The *trxB1*::*erm*/$\Delta trxB2$ double mutant and the $\Delta trxA1$/$\Delta trxA2$/$\Delta trxA3$ triple mutant did not grow on plates when exposed to 1% $O_2$ in contrast to the WT strain, showing the importance of Trx systems in $O_2$ tolerance (Fig 3A and 3B). Furthermore, the $\Delta trxA1$/$\Delta trxA2$ double mutant displayed the same phenotype as the triple mutant at this $O_2$ tension, whereas the two other *trxA* double mutants had a survival rate comparable to the WT strain. These results suggest that TrxA1 and TrxA2 but not TrxA3 are important for survival in the presence of $O_2$. The $O_2$ tolerance of all single mutants was similar to the one of the WT strain, suggesting functional redundancy for TrxA1 and TrxA2 as well as TrxB1 and TrxB2. The phenotypic complementation of double and triple mutants by only one of the above-mentioned partners further confirms this conclusion (S4A Fig). The *trxA3* gene did not complement the *trxA* triple mutant (S4A Fig), supporting the absence of involvement of TrxA3 in $O_2$ tolerance. The same results were obtained in the presence of 0.1% of $O_2$ (S4B Fig).

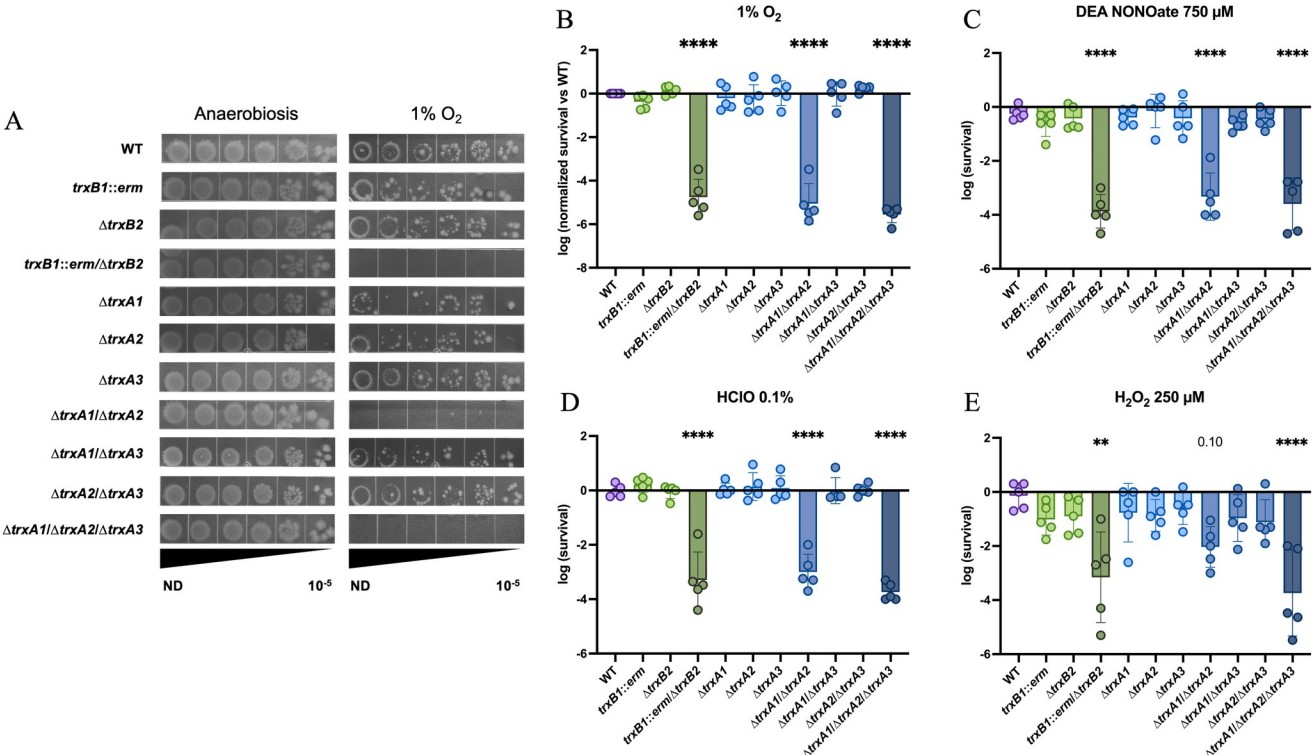

**Fig 3. Trx systems are required for survival to oxidative and nitrosative stresses.** (A, B) Samples were serially diluted, plated in duplicate on TY Taurocholate (Tau) plates and incubated either in anaerobiosis or in hypoxia at 1% $O_2$ for 64 h. Survival was calculated by doing the ratio between CFUs in the last dilution with stress and CFUs in the last dilution without stress. Survival was then normalized as the ratio of the mutant *vs* the WT. (C, D) Samples were serially diluted and plated on TY and (C) TY + DEA NONOate 750 µM or (D) TY + HClO 0.1% and incubated for 24 h. (E) Samples were incubated in glycylglycine buffer in the presence or absence of $H_2O_2$ 250 µM for 30 min, serially diluted and plated on TY plates. Mean and SD of survival are shown. Experiments were performed in 5 biological replicates. For all assays, ordinary one-way ANOVA was performed followed by Dunett's multiple comparison test. The comparisons are made with the WT strain. *: p-value<0.05, ** <0.01, *** <0.001 and **** <0.0001.

We then wanted to study the role of Trx systems in an inflammation context by evaluating the tolerance of *trx* mutants to molecules produced by immune cells. As observed for $O_2$ tolerance, the *trxB1::erm*/Δ*trxB2* double mutant, the Δ*trxA1*/Δ*trxA2* double mutant and the *trxA* triple mutant were more sensitive to DEA NONOate, a NO donor (Fig 3C), to hypochlorite (HClO) (Fig 3D) and to hydrogen peroxide ($H_2O_2$) (Fig 3E), confirming the key role of these four proteins in stress response to inflammation-produced molecules. By contrast, the Δ*trxA1*/Δ*trxA3* and Δ*trxA2*/Δ*trxA3* mutants had a survival rate comparable to the WT strain (Fig 3C–3E), indicating a marginal role of TrxA3 in the survival to oxidative and nitrosative stress, as observed for $O_2$ tolerance. Assays on complemented strains (S4 Fig) validated these observations.

Altogether, these results point out that despite the existence of specific detoxication enzymes for $O_2$ and for oxidative and nitrosative stresses, Trx systems, and more precisely two TrxAs (TrxA1 and TrxA2) and two TrxBs (TrxB1 and TrxB2), are crucial for *C. difficile* survival when exposed to these stresses.

## A supplementary TrxB is part of the stress-response arsenal in some strains of *C. difficile*

An *in silico* investigation for additional Trx partners in NCBI available *C. difficile* assembled genomes led to the identification, in some strains, of a fourth TrxB named TrxB4. In these strains, the *trxB4* gene is located downstream of the *trxA2* gene, which is monocistronic in the

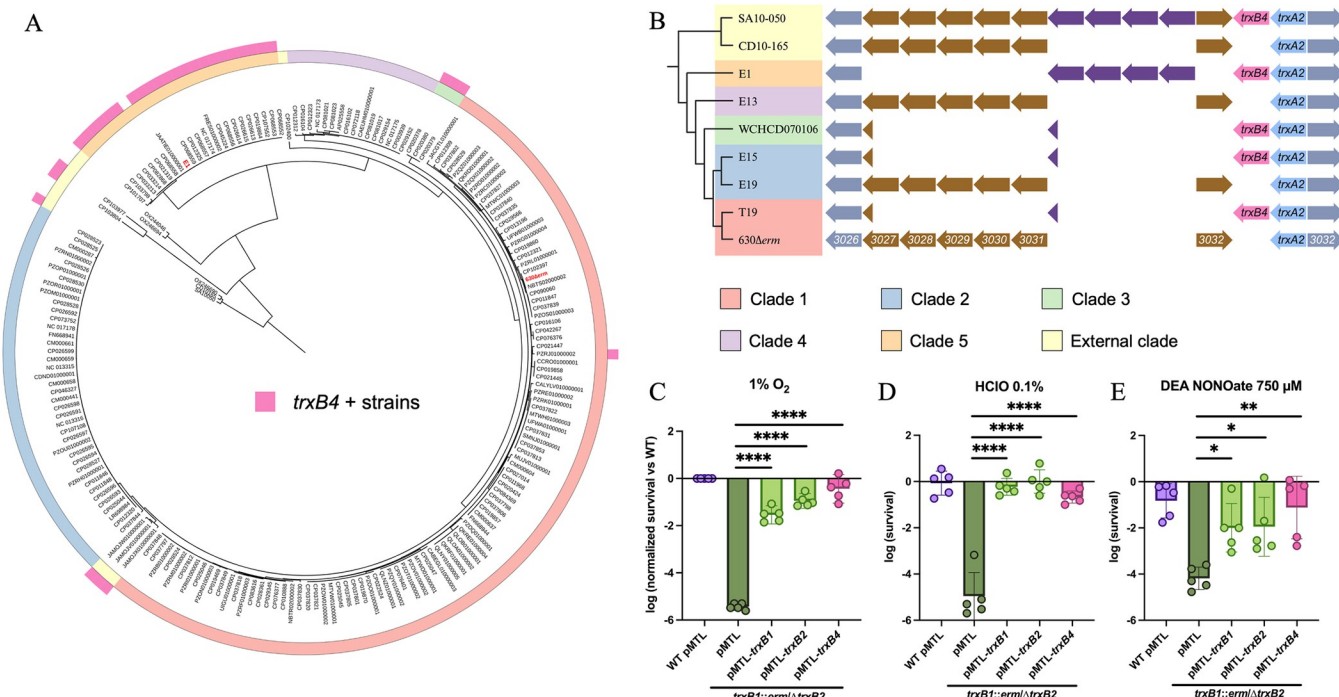

**Fig 4. The fourth TrxB of *C. difficile*.** (A) Repartition of the *trxB4* gene in a data set of 194 genomes. The core genome was used to reconstruct *C. difficile* phylogeny. The presence of *trxB4* was assessed using BLASTN [55] and indicated by pink boxes in the outer layer. Final tree was produced with iTOL [56]. (B) Genetic environment of *trxA2* genes. Sequence of *trxA2* environments were analyzed using MicroScope platform [57]. (C-E) Complementation of the *trxB1/trxB2* double mutant by the *trxB1*, the *trxB2* or the *trxB4* gene. (C) Samples were serially diluted, plated in duplicate on TY Tau plates and incubated either in anaerobiosis or in hypoxia at 1% $O_2$ for 64 h. (D, E) Samples were serially diluted and plated on TY and on (D) TY + HClO 0.1% or (E) TY + DEA NONOate 750 μM. Plates were incubated for 24 h. Experiments were performed in 5 biological replicates. Mean and SD are shown. For stress assays, one-way ANOVA was performed followed by Dunett's multiple comparison test. *: p-value<0.05, ** <0.01, *** <0.001 and **** <0.0001.

630Δ*erm* strain. The *trxB4* gene when present forms an operon with *trxA2*. The *trxB4* gene is widespread in clades 3 and 5 of *C. difficile* but is also found sporadically in some strains of other clades (Fig 4A). The maximum likelihood tree of the *trxB4* gene follows the core-genome *C. difficile* phylogeny (S5A Fig), suggesting an ancestral origin of *trxB4*. We analyzed the genetic environment of *trxA2* in *C. difficile* strains representative of the diversity of the species (Fig 4B). We found that a variable region was delimited by conserved genes, the *trxA2* gene on one side and *CD3026* encoding a hypothetical protein on the other side. In the 630Δ*erm*, this variable region is composed of genes encoding a phosphotransferase system (*CD3027* to *CD3031*) and a PLP-dependent aminotransferase (*CD3032*). These genes are absent in *trxB4*-containing strains, at the exception of the ancestral SA10-050 strain, which is used to root the tree (Fig 4A) [54]. In clade 5 strains carrying *trxB4*, the variable region is composed of an ABC transporter, a membrane protein, a two-component system and *trxB4*. These genes are also present in the SA10-050, confirming that all these genes, including *trxB4*, were ancestral and that several independent gene losses events probably occurred. Consistently, the *trxB4*-containing strains outside of clade 5 harbor scars of the two variable regions, illustrating at least two independent deletions (Fig 4B).

As TrxB1, TrxB4 lacks the NAD(P)H binding motifs and clusters with clostridial FFTRs, strongly suggesting that TrxB4 is a FFTR (S1A–S1B Fig). Given the importance of TrxB1 and TrxA2 in stress-response, we investigated the role of TrxB4 in this adaptive process. We complemented the *trxB1::erm*/Δ*trxB2* double mutant with the *trxB4* gene of the E1 strain [58] expressed under the control of its own promoter, as we show that the σ^A-recognized promoter

located upstream *trxA2* was conserved between the 630Δ*erm* and E1 strains (S5B Fig). The expression of the *trxB4* gene in the *trxB1::erm*/Δ*trxB2* mutant restored a level of O₂-survival comparable to the complementation obtained with the *trxB1* and *trxB2* genes (Fig 4C). The same observations were made with HClO and NO (Fig 4D and 4E), confirming the involvement of TrxB4 in stress-response.

## Two Trx systems are required to cope with disulfide-bond formation induced by some bile salts

Bile acids are inextricably linked to the life cycle of *C. difficile*. Primary bile acids such as cholate (CHO), which are abundant in the CDI context, allow germination [21,23], whereas secondary bile acids such as deoxycholate (DOC) are known to prevent *C. difficile* colonization through growth inhibition [59]. *C. difficile* colonizes the gut at very low concentration of secondary bile acids [23], but faces higher levels during the microbiota recovery phase [60], leading to biofilm formation and/or clearance of the bacteria [61]. Hence, tolerance to bile acids is important for *C. difficile* transmission, persistence, and CDI recurrence.

To determine the implication of Trx systems in bile acids susceptibility, we exposed the *trx* mutants to DOC and CHO (Fig 5A and 5B). As observed for inflammation-related stresses, we found a decrease in survival of multi-mutants of *trxA1*, *trxA2*, *trxB1* and *trxB2* genes,

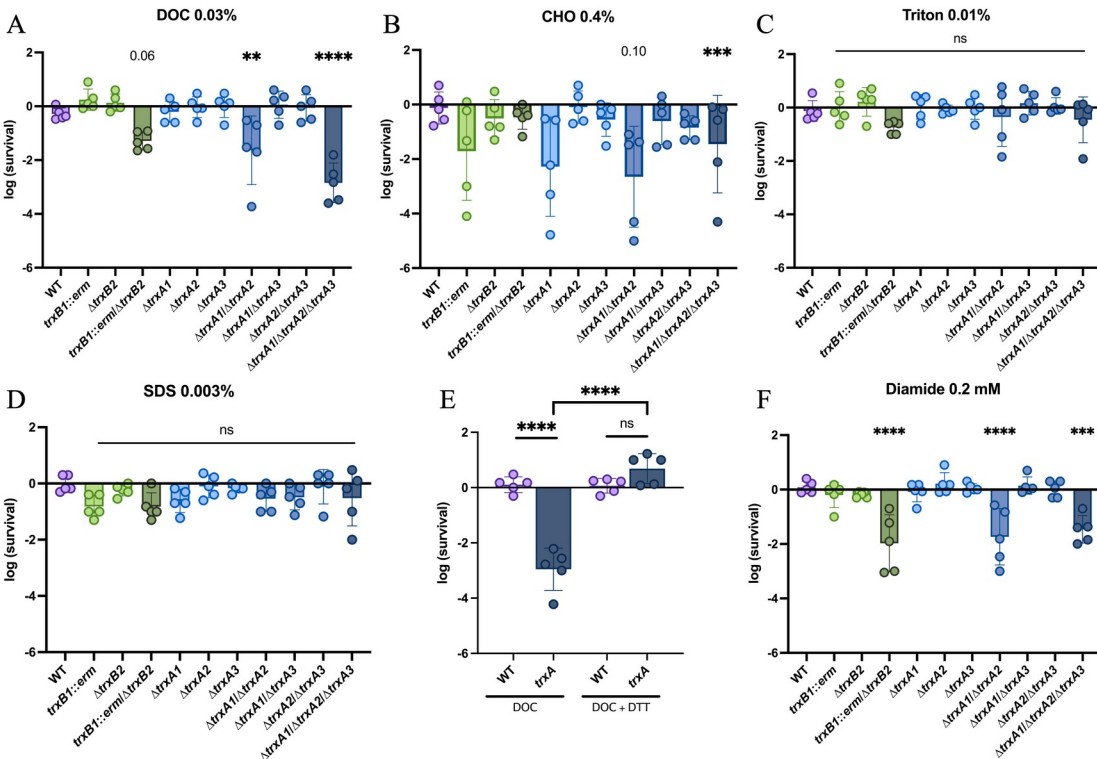

**Fig 5. Trx systems are required to tolerate disulfide-bond forming activity of bile acids.** Samples were serially diluted and plated on TY and on (A) TY + DOC 0.03%, (B) TY + CHO 0.4%, (C) TY + Triton X-100 0.01%, (D) TY + SDS 0.001% or (F) TY + diamide 0.2 mM. The plates were incubated for (F) 24 h or (A-D) 48 h. (E) WT and Δ*trxA1*/Δ*trxA2*/Δ*trxA3* mutant (*trxA*) were serially diluted and plated on TY and on TY containing either DOC 0.03%, DTT 0.1% or DOC 0.03%, and DTT 0.1%. The plates were incubated for 48 h. Survival was calculated by doing the ratio between CFUs in the last dilution with chemical and CFUs in the last dilution without chemical. Experiments were performed in 5 biological replicates. Mean and SD are shown. For all assays, ordinary one-way ANOVA was performed followed by Dunett's multiple comparison test. *: p-value<0.05, ** <0.01, *** <0.001 and **** <0.0001.

suggesting their importance for *C. difficile* survival in presence of DOC, in agreement with a function in stress-response. For CHO, we observed a lesser effect compared to DOC, with only a significant sensitivity for the *trxA* triple mutant. The sensitivity of the complemented multi-mutants was in agreement with these results, with a phenotypic complementation only by *trxA1* or *trxA2* and *trxB1* or *trxB2* (S6A and S6B Fig). Interestingly, this phenotype was not seen when glycodeoxycholate (GlyDOC) was used (S6C Fig), suggesting a specific action of CHO and DOC.

To assess if this increased sensitivity of the mutants was due to the detergent activity of bile acids, we exposed the *trx* mutants to other detergents (Triton X-100 and SDS) (Fig 5C and 5D). We observed no sensitivity of the *trx* mutants at concentrations described to be critical for growth of a detergent-sensitive strain, the Δ*busAA* mutant, used as a control (S6D and S6E Fig) [62]. The *trx* mutants are therefore not sensitive to detergents, suggesting that their bile acids-sensitivity is linked to another physico-chemical property. Bile acids have been described to form disulfide bonds [63], which could explain the observed phenotypes. To test this hypothesis, we exposed the *trxA* triple mutant to DOC in presence of dithiothreitol (DTT), a disulfide-bond reducing agent [64] (Fig 5E). The complete restoration of survival in presence of DTT confirmed that a disulfide bond-dependent mechanism is responsible for the increased susceptibility of *trx* mutants to DOC. This hypothesis is also supported by the fact that DOC-sensitive *trx* mutants were also more sensitive to diamide, a chemical triggering disulfide-bond formation [64] (Figs 5F and S6F). Conjointly, these results show the importance of Trx systems in the repair of disulfide-bonds induced by some bile acids, CHO and DOC.

## The third Trx system is involved in sporulation

Sporulation is a key step of *C. difficile* life cycle. After gut colonization, the formation of spores is triggered by an undetermined signal. The newly-formed spores disseminate in the environment during diarrhea. As this metabolically inactive form is resistant to many stresses including air and acidic pH, it allows the transmission to new hosts. We therefore evaluated the contribution of Trx systems to sporulation. We first evaluated the sporulation capacity of the single mutants (Fig 6A). The Δ*trxA3* mutant showed a significantly lower rate of sporulation, suggesting a role of TrxA3 in this process. The sporulation rate of the Δ*trxA1*/Δ*trxA3* and the Δ*trxA2*/Δ*trxA3* mutants were even lower (Fig 6B), while the Δ*trxA1*/Δ*trxA2* mutant had the same sporulation efficiency than the WT strain. These results suggest a major role for TrxA3 and a slight contribution of the two other TrxAs to sporulation in the absence of TrxA3.

The *trxA3* gene is part of the glycine reductase (*grd*) operon (S1D Fig). As other proteolytic Clostridia, *C. difficile* is able to extract energy from Stickland fermentations [15]. The *grd* operon contains the *grdABCDEX* genes encoding the enzymes involved in glycine reduction, and a Trx system (*trxA3*/*trxB3*) required for the reduction of the selenoenzyme GrdA [65]. To determine if the sporulation defect was associated with the glycine-reductase activity, we studied the sporulation of a Δ*grdAB* mutant (Fig 6C). We observed a sporulation defect, suggesting that GrdAB contributes to the efficiency of spore formation. As glycine is a known co-germinant of *C. difficile* spores [59], we hypothesized that the observed reduction of sporulation efficiency could be due to a reduced consumption of glycine. Addition of glycine in the media decreased the sporulation rate of the WT strain at 24 h (Fig 6D), but at a lower extent than mutations of *grdAB* or *trxA3*, suggesting the existence of another mechanism to explain the observed phenotypes. Altogether, these results show that the third Trx system of *C. difficile*, TrxA3 and likely TrxB3, plays a role in the control of sporulation likely through a GrdAB-dependent mechanism.

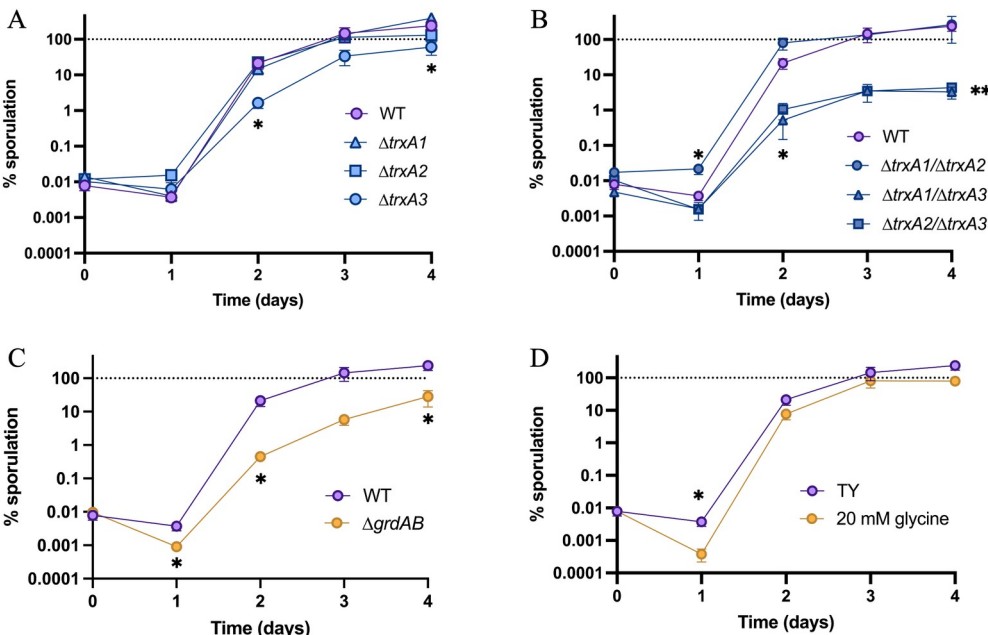

**Fig 6. TrxA3 contributes to spore formation** (A-D) Effect of *trxA* gene inactivation on sporulation. Sporulation rate of WT strain and (A) *trxA* single mutants, (B) *trxA* double mutants and (C) Δ*grdAB* mutant was evaluated daily over 4 days by numeration of total cells and spores by serial dilution and plating on TY + Tau before (total CFUs) and after (spores) ethanol shock. (D) Sporulation rate of WT strain in TY or TY + 20 mM glycine was evaluated. Experiments were performed in 5 biological replicates. Mean of sporulation rate and Standard Error of the Mean (SEM) are shown. For sporulation assays, t-tests were performed comparing the sporulation rate of the condition and the sporulation rate of the WT in TY. *: p-value<0.05, ** <0.01.

## Regulation and role of the spore Trx system

The link between Trx systems and the spores is not restricted to the *grd* operon. Interestingly, TrxA1 and TrxB1 have been detected in the spore proteome of the 630Δ*erm* strain [66]. We used a P$_{trxA1B1}$-*trxA1'*-FAST$^{CD}$ translational fusion to confirm the presence of TrxA1 in the spores. We confirmed that this protein fusion was able to complement the Δ*trxA1*/Δ*trxA2* mutant in a O$_2$-survival experiment (S7A Fig). TrxA1 was detected both in the vegetative cells and in the sporulating cells, mainly in the forespore (Fig 7A). Using a P$_{trxA1B1}$-FAST$^{CD}$ transcriptional fusion (Fig 7B), and the membrane specific marker MTG, we showed that the *trxA1B1* operon was expressed in the vegetative cell (yellow arrows), in the mother cell (pink arrows) and in the forespore (blue arrows). The expression of this fusion was detectable in all the compartments at the stage of asymmetric division. From the engulfment to the maturation of the spore, the expression was significantly increased in the mother cell compared to the vegetative cell, and higher in the forespore compared to the mother cell (Figs 7C and S7B). By contrast, the expression of the P$_{trxA1B1}$-FAST$^{CD}$ fusion was not visible anymore in the phase bright forespore (Fig 7C). In a *sigB::erm* mutant (Fig 7B and 7C), the expression of the fusion was no longer detectable in the vegetative cells consistently with our previous observations (Fig 2D) but also in the mother cell, suggesting that the expression in this compartment was σ$^B$-dependent. Interestingly, the expression of this fusion in the *sigB::erm* mutant was still visible in the forespore, suggesting the existence of an alternative σ$^B$-independent transcription of *trxA1B1* during sporulation.

Four sigma factors are involved in sporulation: σ$^E$ and σ$^K$ in the mother cell and σ$^F$ and σ$^G$ in the forespore [67]. Given the fact that the *trxA1B1* operon was expressed in the forespore in

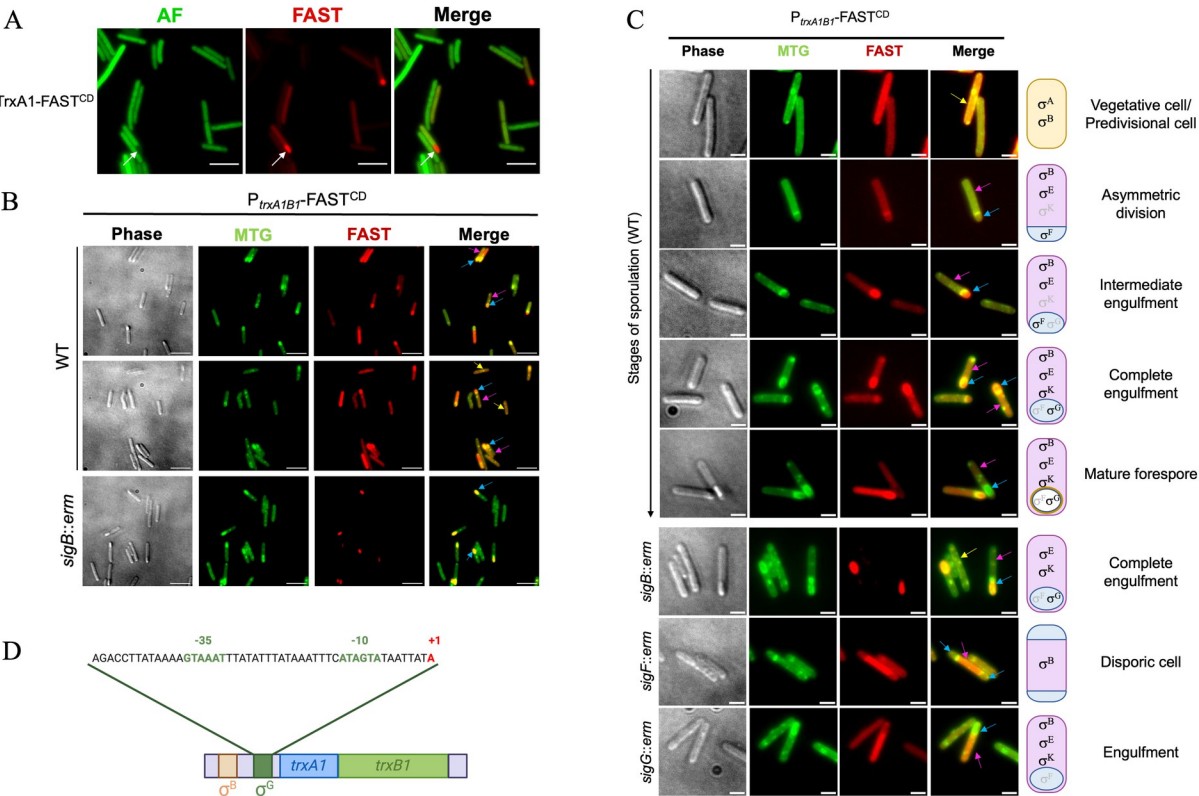

**Fig 7. Regulation of the *trxA1B1* operon during sporulation.** (A) The location of TrxA1 during sporulation was evaluated using the translational P*trxA1B1*-*trxA1'*-FAST^CD fusion. Bacteria were incubated in anaerobiosis for 48 h in sporulation medium and observed at high magnification (100X). Images are overlays of bacterial autofluorescence AF (green) and FAST (red). Scale bars represent 10 μm. White arrows indicate a forespore. (B) Expression of the *trxA1/B1* operon was monitored using the transcriptional P*trxA1B1*-FAST^CD fusion in the WT strain and in the *sigB*::*erm* mutants. Bacteria were cultured for 48 h in anaerobiosis in sporulation medium. Images are overlays of phase contrast, MTG (green) and FAST (red). Scale bars represent 10 μm. Yellow arrows indicate vegetative cells, blue arrows forespores and pink arrows mother cells. (C) Expression of the *trxA1/B1* over the stages of sporulation. The different stages of sporulation of the WT strain are shown. Yellow arrows indicate vegetative cells, blue arrows forespores and pink arrows mother cells. The schematic bacteria indicate the sigma factors present in the compartments over sporulation. The *sigB*::*erm* mutant is shown at the stage of engulfment. The *sigF*::*erm* mutant blocked at the stage of asymmetric division forms disporic cells and the *sigG*::*erm* mutant is blocked at the stage of engulfment. Scale bars represent 3 μm. (D) Promoter identification through 5'RACE using RNA extracted from the *sigB*::*erm* mutant grown in sporulation medium. The TSS (+1) is indicated in red. Upstream this TSS, σ^B and σ^G boxes [46] are represented in orange and green, respectively.

the *sigB*::*erm* mutant, σ^F and σ^G are the best candidates to be the alternative sigma factors that transcribe this operon during sporulation. However, the corresponding promoter was not identified by our first 5'RACE performed with RNA extracted from exponentially growing cells (S2A Fig). We mapped again the TSS using RNA extracted from a *sigB*::*erm* mutant after 24 h of growth in sporulation medium. We were able to identify a new TSS and the consensus sequences of a promoter recognized by σ^G (Figs 7D and S7C and S7D). To confirm this control, we introduced the P*trxA1B1*-FAST^CD transcriptional fusion in the *sigF*::*erm* and the *sigG*::*erm* mutants [67] (Fig 7B and 7C). As σ^F is necessary for *sigG* expression [68], a σ^F-transcribed gene is not expressed in a *sigF*::*erm* mutant but expressed in a *sigG*::*erm* mutant, while a σ^G-transcribed gene is expressed neither in a *sigF*::*erm* nor in a *sigG*::*erm* mutant. In a disporic *sigF*::*erm* mutant cell, the P*trxA1B1*-FAST^CD fusion was expressed only in the mother cell, likely under the control of σ^B, but not in the forespores (Fig 7C). The same observation was made for the *sigG*::*erm* mutant blocked following engulfment completion. These results confirm the σ^G-dependent control of the expression of the *trxA1B1* operon during sporulation in agreement with the second promoter identified.

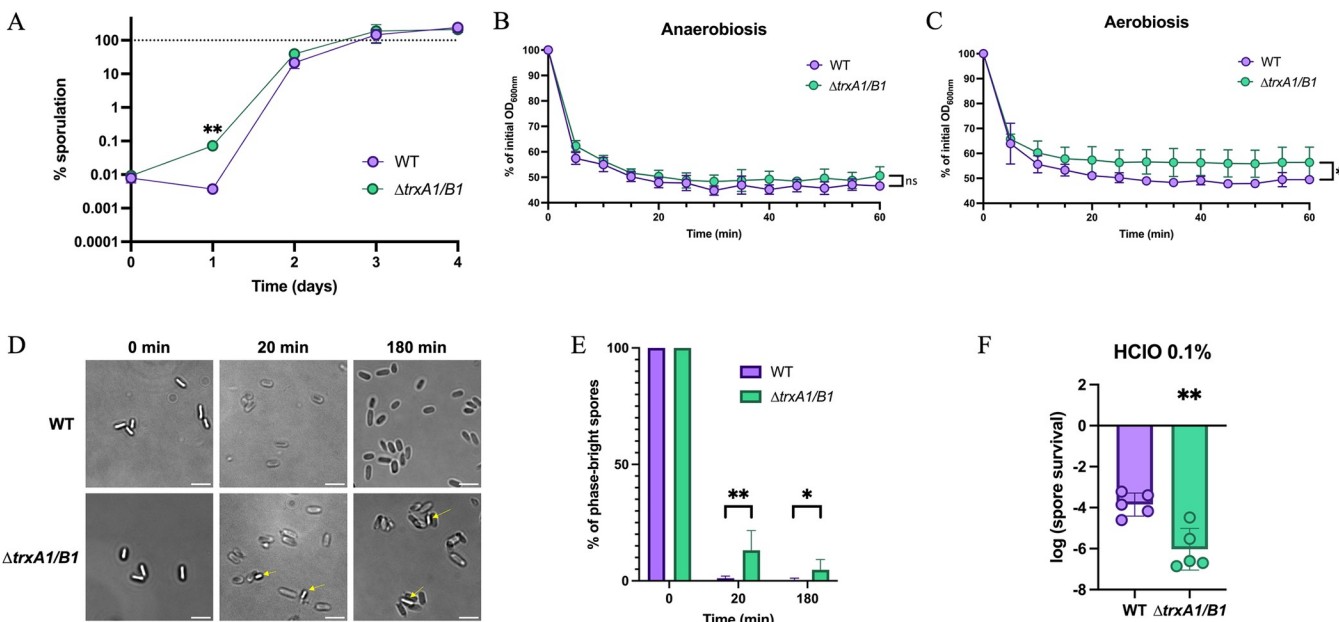

**Fig 8. The role of TrxA1/TrxB1 in spore germination and hypochlorite resistance.** (A) Sporulation rate of the Δ*trxA1*/*trxB1* mutant was evaluated daily over 4 days as described in Fig 6. Mean of sporulation rate and SEM are shown. (B—E) Role of TrxA1/B1 in germination. ~$10^7$ spores of WT strain and of Δ*trxA1/B1* mutant were exposed to 1% taurocholate to induce germination, either (B) in anaerobiosis or (C) in air. $OD_{600nm}$ was monitored every 5 min to evaluate germination. Experiments were performed in 5 replicates with at least 2 independent spore suspensions. Mean and SD are shown. (D) ~$10^7$ spores of WT strain and of Δ*trxA1/B1* mutant were exposed to 1% taurocholate to induce germination. After 0, 20 and 180 min, spores were washed and fixed in 4% PFA before observation under phase contrast microscopy at high magnification (60X). Scale bars represent 5 μm. Yellow arrows indicate bright phase spores. (E) Percentage of bright phase spores were quantified from acquired images of panel C from 900 cells from two independent experiments. Mean and SD are shown. (F) Role of TrxA1/B1 in resistance to HClO treatment. $10^8$ spores were exposed to 0.1% of HClO for 10 min, neutralized with 1% sodium thiosulfate, washed and diluted and spotted on TY Tau plates for numeration. Survival was estimated by calculating the ratio of CFUs between treated and non-treated spores. Experiments were performed in 5 biological replicates. Mean and SD are shown. For sporulation assays, t-tests were performed comparing the sporulation rate of the WT and the mutant. For germination assays, two-way ANOVA were performed. For comparison of the proportion of bright phase spores, multiple unpaired t tests were performed. For sporicidal assays, unpaired t-tests were performed. *: p-value<0.05, ** <0.01.

To understand the function of this Trx system in the spore, we first performed a sporulation assay using a Δ*trxA1/B1* double mutant (Fig 8A). This mutant has a similar sporulation rate as the WT strain, except after 24 h where the percentage of spores was significantly higher. This quicker sporulation is not due to a difference in growth (S7E Fig). The TrxA1/B1 system might thus contribute to delay sporulation but is clearly not required for sporulation. We then performed a germination assay using spores of both the WT and the Δ*trxA1/B1* mutant (Fig 8B). The germination was followed in anaerobiosis and in air (Fig 8B–8C) through measurement of the decrease of $OD_{600nm}$ of a spore suspension following addition of 1% taurocholate. The WT and Δ*trxA1/B1* mutant spores germinated similarly in anaerobiosis (Fig 8B), but the total $OD_{600nm}$ reduction was significantly lower in the mutant in aerobiosis (Fig 8C), suggesting a lower efficiency of germination. We also evaluated the germination in aerobiosis after 20 and 180 min under the microscope (Fig 8D and 8E). After 20 min, all the spores of the WT strain were phase-dark, while a subpopulation of approximately 15% of spores of the Δ*trxA1/B1* mutant remained phase-bright. After 180 min, this subpopulation only represented 5% of spores, which was still significantly higher than the WT strain. In addition, the reduced decrease of $OD_{600nm}$ observed in aerobiosis for the Δ*trxA1/B1* was still observed when DTT was added (S7F Fig). This result suggests that the reduced efficiency of germination is not due only to the disulfide bond reduction activity of Trx systems. Moreover, as a phenotype was only visible in aerobiosis, it seems that this germination defect is due to an increased sensitivity

to oxidative stress. We also performed an outgrowth assay in anaerobiosis of the mutant, but we observed no difference with the WT strain (S7G Fig). However, as this assay requires anaerobiosis to observe growth, we are not able to conclude if the germination defect observed in aerobiosis would lead to an outgrowth defect in similar conditions.

To validate the hypothesis of a link with oxidative stress and knowing the impact of these TrxA1/B1 in oxidative stress tolerance (Fig 3), we tested the viability of the spores when exposed to a sporicidal molecule, HClO (Fig 8F). After 10 min of exposure at 0.1% HClO, we found that the survival of Δ*trxA1/B1* spores was more impacted than the one of the WT spores with a 33-fold decrease of spore survival. These results confirm the role of the TrxA1B1 system in the protection of the spore against oxidative stress and suggest a role during germination.

## Discussion

In this work, we used *C. difficile* as a model to understand the multiplicity of Trx systems in Clostridia and more broadly in bacteria. This representative of Clostridia can be genetically modified and is a major healthcare problem. Digging in its atypical but well-characterized lifestyle is thus of great interest, both for the understanding of the complexity of Trx systems and the better characterization of the physiology of the pathogen.

Two of the three Trx systems of *C. difficile* are involved in stress response while the third one is involved in glycine catabolism and sporulation. TrxA1, TrxB1, TrxA2 and TrxB2 play a crucial role in the response to infection-related stresses such as bile acids, $O_2$, ROS, HClO and NO. These Trx systems contribute to a first line of defense against molecules produced by immune cells, especially neutrophils, which are massively recruited during CDI [32]. The absence of phenotypes for single mutants or for double mutants complemented by one of the copies indicates a functional redundancy between TrxA1 and TrxA2 on one hand and between TrxB1 and TrxB2 on the other hand. Whether each TrxB can reduce both TrxA1 and TrxA2 or whether specialized pairs exist remain to be determined. The TrxA1 protein and the TrxB1 FFTR, that are encoded by the same operon and detected both in vegetative cells and in spores [66], very likely form a first stress-dedicated pair. It is also noteworthy that the *trxA2* gene forms an operon with *trxB4*, which encodes a FFTR in strains containing this fourth copy, suggesting that TrxA2 might function with FFTRs. In strain 630Δ*erm*, both the *trxA1B1* operon and the *trxA2* gene are induced during long term exposure to 1% $O_2$, but these genes are transcribed using different σ factors. These results suggest the existence of a common still uncharacterized regulator.

The spore-associated TrxA1/B1 system is important to protect the spore from oxidants including HClO, which is produced by immune cells but also used as a disinfectant in hospital to eradicate spores [69]. The TrxA1/B1 system probably also contributes to the protection of the germinating cells in the small intestine where the $O_2$ tension is around 4–5% [47,70]. TrxB1 uses a ferredoxin instead of NAD(P)H as substrate [11]. As spores are metabolically dormant, NAD(P)H is probably not renewed. It is noteworthy that ferredoxins are present in the spore, as well as enzymes that could compose their renewal system, the pyruvate-ferredoxin oxidoreductase (CD2682) and an iron-only hydrogenase (CD3258) [66]. Using FFTRs may therefore allow *C. difficile*, and likely other Clostridia, to have a functional TrxB in the spore, even in absence of NAD(P)H. Seven ferredoxins, potential partners of the FFTR, are encoded in the genome of *C. difficile*. Among them, CD0115, CD1595.1 and CD3605.1 are present in the spore, strongly suggesting that at least one acts as a partner of TrxB1. *CD3605.1* is controlled by σ[B] and induced in the presence of 1% $O_2$ (S2B and S2C Fig), following the same expression pattern as the *trxA1B1* operon. Interestingly, *CD1595.1* is less expressed in a *sigF* and a *sigG* mutant in transcriptome [71] and transcribed from a promoter recognized by

σ$^G$ [46], sharing this control with *trxA1B1* during sporulation. These ferredoxins, which are detected in the proteome of the vegetative cells for CD3605.1 [72] and of the spores for CD3605.1 and CD1595.1 are good candidates as main partners of TrxB1.

TrxA3/TrxB3 are encoded by genes belonging to the *grd* operon. The inactivation of either *grdAB* or *trxA3* leads to a reduced sporulation efficiency, consistently with a recent study about the role of GrdAB in sporulation [73]. This shared phenotype suggests that TrxA3 and likely TrxB3 correspond to the Trx system dedicated to the glycine reductase and the reduction of GrdA. GrdAB and its Trx system would therefore allow *C. difficile* to consume glycine, a spore co-germinant [59]. The gene content of the *grd* operon, including the presence of *trxA3/B3*, is conserved in other proteolytic Clostridia including *Clostridium sticklandii*, *Paraclostridium bifermentans*, *Paeniclostridium sordelli* and *Clostridium botulinum*, even if several genetic rearrangements are observed (S1D Fig). This metabolism allow the extraction of energy from glycine by Stickland fermentation [15,65,74] and is crucial for niche colonization, as interspecies competition for glycine has been shown to be a determinant of *C. difficile* germination and colonization [75]. In addition to the systematic presence of *trxA3/B3* in the *grd* operon, TrxA3 contains an atypical [G/S]C[V/E]PC active site described in the *grd*-associated TrxA of other proteolytic Clostridia [44], while TrxA1 and TrxA2 contain a classical TrxA active center motif WCGPC [5] (S1C Fig). This *grd*-associated TrxA has been shown to function with Clostridial NTR but not with *E. coli* NTR, suggesting that TrxB3 might be dedicated only to the atypical TrxA3 protein and supporting the dedication of the TrxA3/B3 system to the reduction of GrdAB. The difference in the CXXC catalytic motif might be important to discriminate TrxA targets. In *H. pylori*, Trx1 and Trx2 are both TrxAs, but only the classical Trx1 with the WCPGC motif is able to reduce AhpC, a peroxiredoxin [76]. TrxA1 and TrxA2 have probably a broader range of partners than TrxA3. These diverse targets could contribute to the role of TrxA1 and TrxA2 in stress-response. Indeed, Trx systems have both a direct and indirect effect in stress-survival. The thiol-disulfide exchange reaction is directly required for thiol repair following stress exposure. Indeed, DOC, diamide and HClO trigger a disulfide-stress [77], ROS oxidize SH groups to disulfide or sulfenic acids and NO leads to S-nitrosylation. An indirect role would be mediated by detoxication or repair enzymes such as methionine sulfoxide reductase (MsrAB) and peroxiredoxins [thiol-peroxidase (Tpx) or bacterioferritin comigratory protein (Bcp)] [78–80] that require Trx and/or Grx systems for their activity. In bacteria lacking Grx and AhpF such as *H. pylori*, AhpC depends only on Trx for its activity [76]. In *C. difficile*, MsrAB and Bcp, although uncharacterized, are present. The *msrAB* and *bcp* genes are members of the σ$^B$ regulon [33], in agreement with the increased sensitivity of the *sigB* mutant to oxidation [33,50]. This observation also indicates a coordinated regulation of a large set of genes encoding enzymes involved in ROS detoxication and damage repair. Another potential Trx target is CotE, a bifunctional spore-coat protein with chitinase and peroxiredoxin domains [81]. This second domain might be reduced by TrxA1/B1 in the spore. Finally, Trx systems also contribute to the activity and the recycling of proteins involved in central metabolism such as the ribonucleoside-di-phosphate reductase (CD2994-CD2995) [82] and GrdAB.

The specificity of TrxA targets could explain, at least partly, the multiplicity of Trx systems as observed for other detoxication enzymes. *C. difficile* encodes indeed many redundant detoxication enzymes, *e.g.* three catalases, four peroxidases, two flavodiiron proteins and two reverse rurbrerythrins [47]. The evolutionary origin of these multiplications remains unexplored. The Trx systems seem to be ancestral as they are conserved among *C. difficile* strains, at the exception of *trxB4*. TrxB2, a NTR, which is present in the vegetative cell, is involved in stress-response and is encoded by a monocistronic gene, would correspond to the ubiquitous bacterial TrxB copy able to reduce various TrxAs as observed in other bacteria [7]. The *trxA2-trxB4* operon could be a duplication of the *trxA1-trxB1* operon, as both TrxBs are FFTR with a high

level of similarity (78% of similarity at the protein level). This is also true for TrxA, as TrxA1 and TrxA2 are closer (63% similarity) than TrxA3 (33 and 35% of similarity with TrxA1 and TrxA2, respectively). The multiplicity of stress-response genes could be associated with the lifestyle of *C. difficile*, a gut resident exposed to various $O_2$ tensions, oxidative and nitrosative stresses [47]. Such multiplicity has already been observed in other gut anaerobes such as *Bacteroides* [83], with notably six TrxAs in *Bacteroides fragilis* [9]. Several TrxBs are also present in Fusobacterium, another gut anaerobic bacterium (Fig 1A). Conversely, only one TrxB, an NTR, is present in the aerobic members of the Firmicutes, Bacilli.

The multiplication of Trx systems has also been described in other organisms such as Cyanobacteria, with the existence of several TrxAs and TrxBs in the same organism [84]. Cyanobacteria are aerobic bacteria exposed to strong oxidative stress due to light exposure, aerobic photosynthesis and respiration [85]. Some Cyanobacteria fix nitrogen in condition of anoxia [86]. In these nitrogen-fixating organisms, TrxBs are key sensors of the redox state that reroute metabolism either towards respiration or nitrogen fixation [84]. Some TrxBs are indeed specific of dedicated compartments, either temporal with the day/night cycle, or spatial with heterocysts, which are specialized cells performing nitrogen fixation in filamentous Cyanobacteria [84,87]. Parallels could be drawn between the cell compartmentation and differentiation of Cyanobacteria and the sporulation of Clostridia. Heterocysts are anoxic cells in aerobic organisms, while spores of anaerobes tolerate $O_2$ and ROS at significantly higher levels than vegetative cells [27]. Multiplication and evolution of adapted Trx systems would therefore meet the specific needs of compartmentation and differentiation. This could be extended beyond bacteria, as dedicated Trx systems are maintained specifically in chloroplasts and mitochondria [88,89].

## Methods

### Phylogenetic analysis of bacterial Trx systems

A dataset of 349 diversity-representative bacteria was used as the database for all bacteria [38]. A dataset of 67 bacteria was used as the Firmicutes database. BLASTP [39] was used using TrxB sequence from *E. coli* (accession number P0A9P5—TRXB_ECO57) and TrxB1 sequence from *C. difficile*. Presence of the NAD(P)H binding motifs (GxGxxA and HRRxxxR motifs [11]) were investigated for detection of NTR, while non-NTR were defined by the absence of the motifs. Schematic phylogeny was made following the one in Witwinoski *et al.* [40] for all bacteria and following Taib *et al.* [43] for Firmicutes. The number of TrxAs and TrxBs was reported following BLASTP results.

### Bacterial strains and culture media

*C. difficile* strains and plasmids used in this study are listed in S1 and S2 Tables. *C. difficile* strains were grown anaerobically (5% $H_2$, 5% $CO_2$, 90% $N_2$) in TY (Bacto tryptone 30 g.L$^{-1}$, yeast extract 20 g.L$^{-1}$, pH 7.4), in Brain Heart Infusion (BHI; Difco) or in sporulation medium [90]. For solid media, agar was added to a final concentration of 17 g.L$^{-1}$. When necessary, thiamphenicol (Tm, 15 µg.mL$^{-1}$), erythromycin (Erm, 2.5 µg.mL$^{-1}$) and cefoxitin (Cef, 25 µg.mL$^{-1}$) were added to *C. difficile* culture. *E. coli* strains were grown in LB broth. When indicated, ampicillin (Amp, 100 µg. mL$^{-1}$) and chloramphenicol (Cm, 15 µg.mL$^{-1}$) were added to the culture medium. When indicated, the spore germinant taurocholate (Tau) was added in plates at 0.05%.

### Construction of *C. difficile* mutant strains

All primers used in this study are listed in S3 Table. The *trxB1::erm* mutant was obtained by using the ClosTron gene knockout system as previously described [91]. The PCR product

generated by overlap extension that allows intron retargeting to *trxB1* was cloned between the HindIII and BsrG1 sites of pMTL007 to obtain pDIA6190. The Δ*trxB2*, Δ*trxA1*, Δ*trxA2*, Δ*trxA3*, Δ*trxA1/B1* and Δ*grdAB* knock-out mutants were obtained by using the allele chromosomic exchange method [92]. Briefly, PCR were performed to amplify 1 kb fragments located upstream and downstream of the targeted genes. Using the Gibson Assembly Master Mix (Biolabs), purified PCR fragments were cloned into the pMSR plasmid [92] linearized by inverse PCR and treated by DpnI. The sequence of the resulting plasmids was verified by sequencing. These plasmids (S1 Table) introduced in the HB101/RP4 *E. coli* strain were then transferred by conjugation into *C. difficile* strains. Transconjugants were selected on BHI plates supplemented with Tm and *C. difficile* selective supplement (SR0096, Oxoid). Single crossover integrants were identified by a faster growth on the selective plates. These clones were restreaked into new BHI plates containing Cef and Tm. They were then restreaked on BHI plate containing 200 ng.mL$^{-1}$ of aTc. This compound allowed the expression of the *CD2517.1* toxin gene cloned under the control of the $P_{tet}$ promoter and the selection of a second crossover event [92]. The resulting clones were checked by PCR for the expected deletion. Steps were repeated in each different mutant to generate *C. difficile* multi-mutants.

## Complementation of the different mutants

For the construction of the complementation plasmids, different strategies were used (S3 Table). To complement the mutants with *trxA1* or *trxB1*, a fragment containing their promoter region and the *trxA1* plus *trxB1* genes (IMV1183/IMV1151) was first cloned between the XhoI and BamHI sites of pMTL84121 giving pDIA7025. The plasmid pDIA7025 (pMTL84121-P-*trxA1*-*trxB1*) was then linearized by inverse PCR to delete either *trxA1* (IMV1200/IMV1201) or *trxB1* (IMV1331/CM13) to give pDIA7042 or pDIA7129, respectively. The *trxA2* gene or the *trxB2* gene with their promoter regions were amplified by PCR using CA24/CA25 (*trxA2*) or IMV1202/IMV1203 (*trxB2*) and cloned into pMTL84121 linearized by inverse PCR with the oligonucleotides CM13 and IMV993 using the Gibson Assembly method giving pDIA7122 and pDIA7050. A PCR fragment corresponding to the *grd* operon promoter region and the *grdX*-*trxB3*-*trxA3* genes was amplified using IMV1235/CA43. The PCR product was cloned by the Gibson Assembly method into pMTL84121 to produce pDIA7162. The *grdX* and *trxB3* genes were deleted from plasmid pDIA7162 by inverse PCR using IMV1300/IMV1389 giving pDIA7164. For the plasmid containing *trxB4*, a PCR fragment corresponding to the promoter region and the *trxA2* and *trxB4* genes was first obtained using CA26/CA27 and DNA from the E1 strain as template. The fragment was cloned into pMTL84121 by the Gibson Assembly method to give pDIA7113. Inverse PCR was then performed with CA75/CA76 to remove the *trxA2* gene giving pDIA7156.

All the plasmids were verified by sequencing, introduced into HB101/RP4 *E. coli* strain and then transferred into *C. difficile* strains by conjugation. Transconjugants were selected on BHI plates supplemented with Tm and *C. difficile* selective supplement. The presence of the cloned gene was then verified by PCR.

## RNA extraction, qRT-PCR and 5'RACE

Cultures of the WT strain were inoculated in TY at $OD_{600nm}$ 0.05 and incubated for 24 h in anaerobiosis or with 1% $O_2$ (BugBox M from Baker Ruskinn). To ensure proper gas diffusion, cultures were made with 2 ml in 6-well plates. Cultures of the WT strain and the *sigB*::*erm* mutant were harvested after 4.5 h of culture in TY. Pellets were resuspended in the RNApro solution (MP Biomedicals) and RNA was extracted using the FastRNA Pro Blue Kit (MP Biomedicals). cDNAs synthesis and real-time quantitative PCR were performed as previously

described [51,93]. In each sample, the quantity of cDNAs of a gene was normalized to the quantity of cDNAs of the *gyrA* gene. The relative change in gene expression was recorded as the ratio of normalized target concentrations (the threshold cycle ΔΔCt method) [94]. Experiment was performed in at least 6 biological replicates.

To determine the transcription initiation site of *trxA1* and *trxB2*, the 5' RACE (Rapid Amplification of cDNA Ends, Invitrogen kit) technique was used. Using SuperScript II reverse transcriptase in the buffer provided (20 mM Tris-HCl (pH 8.4), 50 mM KCl, MgCl2 2.5 mM, dNTP 0.4 mM, DTT 0.01 mM), a single-strand cDNA was synthesized from a site internal to the unknown 5' end of the mRNA using an oligonucleotide specific of each gene. For the *trxA1* gene, we used either mRNA extracted from the WT strain after 5 h of growth in TY or from the *sigB* mutant after 24 h of growth in sporulation medium. The matrix mRNA was eliminated by treatment with H and T1 RNAses. A polyC tail was then added to the 3' end of the cDNA *via* the terminal deoxynucleotidyl transferase. A PCR was then performed using a primer complementary to the polyC tail (containing an arbitrary sequence in 5') and another primer specific to each cDNA. A second amplification of the obtained PCR product is performed with oligonucleotides specific for the start of each gene and a primer hybridizing with the arbitrary sequence located upstream of the polyC (Abridged Anchor Primer APP). The PCR products were cloned into the plasmid pGEMT easy (Promega). For each gene, 4 to 8 plasmids corresponding to white colony-forming transformants were extracted and sequenced to determine the TSS.

## Fusion with the FAST[CD] reporter system

To develop a new fluorescent reporter system for use in *C. difficile*, we turned to the FAST-tag reporter, a 14-kDa monomeric protein, engineered from the photoactive yellow protein, that interacts rapidly and reversibly with 4-hydroxybenzylidene rhodamine derivatives [48,49]. We designed a synthetic FAST cassette with a codon usage optimized for *C. difficile* (Integrated DNA Technology), which we termed FAST[CD]. This cassette was cloned into pFT47 [67] by replacement of the SNAP gene by the FAST[CD] gene to give pDIA7190. This plasmid was then linearized by inverse PCR using IMV1103 and either IMV1440 (transcriptional fusions) or IMV1498 (translational fusion). We amplified by PCR a 210 bp DNA fragment corresponding to the promoter region upstream of the *trxA1B1* genes using primers IMV1105/IMV1443 (S2 Table). Using the Gibson Assembly method, we inserted this promoter region into pDIA7190 to obtain pDIA7194 carrying a transcriptional P*trxA1B1*-FAST[CD] fusion. The promoter region and the complete sequence of *trxA1* without its stop codon using IMV1105/IMV1499 was also amplified and cloned into pDIA7190 to obtain pDIA7236 carrying a translational P*trxA1B1*-*trxA1'*-FAST[CD] fusion. These plasmids were introduced into *E. coli* HB101 (RP4) and then transferred to *C. difficile* 630Δ*erm*, *sigB::erm*, *sigF::erm* or *sigG::erm* strain by conjugation. Transconjugants were selected on BHI agar plates containing Tm and Cfx.

## Fluorescence microscopy and image analysis

For FAST labelling, cells were harvested after 24 h of growth in anaerobiosis or at 1% of $O_2$ in TY or after 48 h in sporulation medium in anaerobiosis. Cells were washed once (PBS), fixed in 4% paraformaldehyde (PFA) in PBS for 20 min at room temperature and gently washed 4 times (PBS). Cells were then resuspended in 100 μl in PBS and the TF-coral substrate (Twinkle Factory) was added at a final concentration of 5 μM. The membrane dye Mitotracker Green (MTG) (0.5 μg/ml, ThermoFisher) was added before fixation and the suspension was incubated at room temperature for 15 minutes in the dark.

For phase-contrast and fluorescence microscopy, 5 μL of vegetative and/or sporulating cells were mounted on 1.7% agarose-coated glass slides to keep the cells immobilized and to obtain sharper images. The images were taken with exposure times 600 ms for FAST and FITC. Vegetative cells were observed using a Nikon Eclipse TI-E microscope 60x Objective and captured with a CoolSNAP HQ2 Camera. For the spores, cells were observed using a Delta Vision Elite microscope equipped with an Olympus 100x objective and captured with sCMOS camera. The images were analyzed using ImageJ. Acquisition parameters were similar for all samples of an experiment. Average of FAST fluorescence intensity consists of fluorescence intensity of at least 600 bacteria from two independent experiments from 3 different microscopic fields.

## Growth curves, survival and sporulation assays

Overday cultures of *C. difficile* strains were inoculated by a 1:50 dilution in TY medium. After 5 h of growth, new bacterial suspensions were prepared at an $OD_{600nm}$ of 0.05 in TY and 100 μl were used in 5 technical replicates for preparation of 96 wells plates that were sealed with a gas-impermeable adhesive film (MicroAmp Optical Adhesive Film, Applied Biosystems). $OD_{600nm}$ was monitored every hour at 37°C for 24 h by using a GloMax Explorer plate-reader (Promega).

For sporulation assays, 5 ml bacterial suspensions were prepared from an overday culture at an $OD_{600nm}$ of 0.05 in TY. At 0, 24, 48, 72 and 96 h, the number of total cells was determined by serial dilution and plating on TY Tau. The spore number was determined by replating the dilution after an ethanol shock (volume 1:1 of absolute ethanol) of 1h. Sporulation rate was estimated by calculating the ratio of spores to total cells over time. Total cells data from 0, 24 and 48 h were used for survival curves. Experiments were performed in 5 biological replicates.

## O$_2$-tolerance assays

Overday cultures in TY Tau were used to prepare bacterial suspensions at an $OD_{600nm}$ of 0.5. After serial dilution (non-diluted to $10^{-5}$), 5 μl of each dilution were spotted on TY Tau plates. Plates prepared in duplicate were incubated at 37°C either in anaerobia or in the presence of 0.1% or 1% of $O_2$ for 64 h. The last dilution allowing growth after incubation was recorded and the CFUs of this dilution were counted. Survival was calculated by doing the ratio between CFUs in the last dilution in hypoxia and CFUs in the last dilution in anaerobiosis. The results were then normalized by doing the ratio between the results obtained for a mutant with the results of the reference strain, 630Δ*erm* or 630Δ*erm* pMTL84121 for plasmid-carrying strains. Experiments were performed in 5 biological replicates.

## Stress tolerance assays

NO, HClO, DOC, CHO, SDS, Triton X-100 and diamide stress assays were performed in TY by using the serial dilution protocol as indicated in the $O_2$-tolerance section. Dilutions of a suspension at an $OD_{600nm}$ of 0.5 were plated on TY plate and in TY plate containing either 750 μM of DEA NONOate (Sigma-Aldrich), 0.1% of NaClO (Sigma-Aldrich), 0.03% (725 μM) of DOC (Sigma-Aldrich), 0.4% (9.3 mM) of CHO (Sigma-Aldrich), 0.1% (2.2 mM) of GlyDOC (Sigma-Aldrich), 0.01% of Triton X-100 (Merck), 0.003% of SDS (Euromedex) or 0.2 mM of diamide (Sigma-Aldrich). When indicated, DTT (Sigma-Aldrich) was added at 0.1%. For each plate, the last dilution allowing growth was recorded after incubation at 37°C for 24 h (NO, HClO, diamide) or 48 h (DOC, CHO, SDS, triton). The survival was evaluated by doing the ratio of the CFUs in presence of the stress on the CFUs on the control plate. For $H_2O_2$, overday cultures were used to prepare two bacterial suspensions per strain at an $OD_{600nm}$ of 0.5 in glycyl-glycine buffer (glycylglycine 50 mM, glucose 0.2%, pH 8). $H_2O_2$ (Honeywell) was added in

one of the suspensions to a final concentration of 250 µM. After 30 minutes, the suspensions were serially diluted and plated on TY plate and survival was calculated as mentioned above. Experiments were performed in 5 biological replicates.

### Spore production, germination, outgrowth and sporicidal assays

Spore suspensions were prepared as previously described [95]. Briefly, 200 µl from overnight cultures of *C. difficile* strains were plated on SMC agar plates and were incubated at 37˚C for 7 days. Spores scraped off with water were incubated for 7 days at 4˚C. Cell fragments and spores were separated by centrifugation using a HistoDenz (Sigma-Aldrich) gradient [96]. Spores were enumerated by CFU calculation on TY Tau.

Spore germination was monitored by $OD_{600nm}$ as previously described [95]. Briefly, a spore suspension was resuspended in BHI and exposed to 1% Tau, and the $OD_{600nm}$ was measured every 5 min for 1 h, either in anaerobiosis or in air, with or without DTT at 0.1%. Experiments were performed in 5 independent replicates, with at least 2 independent spore preparations. For evaluation under the microscope, a spore suspension was resuspended in BHI and exposed to 1% Tau. After 0, 20 and 180 min, spores were washed and fixed in 4% PFA. Spores were then observed under phase contrast microscopy under 60x magnification using a Nikon Eclipse TI-E microscope 60x Objective and captured with a CoolSNAP HQ2 Camera. The images were analyzed using ImageJ. Percentage of bright phase spores were quantified from 900 cells from two independent experiments from three different microscopic fields.

Outgrowth was monitored by $OD_{600nm}$ as previously described [97]. A spore suspension was resuspended in BHI and exposed to 1% Tau, and the $OD_{600nm}$ was measured every 10 min for 1 h followed by every 30 min for 10 h. $OD_{600nm}$ was normalized by the initial $OD_{600nm}$. Experiments were performed in 5 independent replicates, with at least 2 independent spore preparations.

The sporicidal assay was performed as previously described [98]. Spore suspensions were exposed for 10 min to 0.1% HClO or to water. 1 volume of 1% sodium thiosulfate was then added to neutralize HClO, and the spores were washed twice in water. After resuspension, spores were serial diluted and spotted on TY Tau for quantification. Survival corresponds to the ratio between treated spores on untreated spores. Experiments were performed in 5 independent replicates, with at least 2 independent spore preparations.

### *C. difficile* core genome phylogenetic tree

Using Bio.Entrez package of Biopython [99] we retrieved all the accession numbers for *C. difficile* available in GenBank database [100] (retrieved on 2022-12-27), with the following query: ("*Clostridioides difficile*"[Organism] OR "*Clostridium difficile*"[Organism]) AND ("4000000"[SLEN]: "10000000"[SLEN]). 2251 accession numbers were fetched. For those of them that had a sequence associated and whose molecular type was "genomic DNA", we retrieved their sequences and metadata. We thus obtained a dataset of 212 *C. difficile* genomes. We added three custom genomes to this data set (E1, CD10165 and SA10050) [54,58], obtaining a data set of 215 sequences. Redundant strains were removed, leading to the final data set of 194 genomes. We annotated these genomes in our data set, extracted their core genome and aligned it with PanACoTA [101] (v1.3.1, the gene annotation was performed with Prodigal [102]). We then reconstructed a phylogenetic tree for the core genome with IQ-TREE 2 [103] (v2.2.2, with model selection [104]: -m TEST, partition [105] by 1st, 2nd and 3rd codon positions, and 1000 ultrafast bootstraps [106]). In the resulting tree, we collapsed the non-informative branches (of less than 1/2 mutations) with gotree [107]. The final figure was produced with iTOL [56].

## Gene-specific phylogenetic trees

We detected the orthologs of the four *trxB* genes (*trxB1*, *trxB2*, *trxB3*, and *trxB4*) in the 194 genomes of our data set with BLASTN [55]. For each of the four genes we created a multiple sequence alignment with Clustal Omega [108] available in SeaView [109] (v5.0.5). Each alignment was manually curated. We reconstructed the phylogenetic trees for each gene with the same procedure as for the core tree. We then compared them to the core tree with ETE3 toolkit [110]: the trees were highly compatible, with 96–98% of their branches present in the core-genome tree. The final figure was produced with Dendroscope [111].

## Protein alignment, distance tree and synteny analysis

TrxA and TrxB sequences from diverse bacteria were aligned with MAFFT software [112] using the Auto Strategy option of the software. Alignment was visualized on Jalview software [113]. For TrxBs, distance tree was obtained using neighbor joining method with a JTT substitution model. TrxB from *Desulfovibrio vulgaris* [114] was used as outgroup to root the tree. Final tree was produced with iTOL [56]. For synteny analysis, *grd* operons and *trxA2* genetic context from Clostridia and *C. difficile* strains were analyzed using the MicroScope website [57].

## Statistical analysis and data presentation

For survival assays, multiple unpaired t-tests were performed. For qPCRs, one sample t-tests were performed with comparison of the fold change to 1. For microscopy, Kruskal-Wallis tests were performed followed by Dunn's multiple comparison test. For stress assays, one-way ANOVA were performed. When significant, multiple comparison was performed using Dunett's multiple comparison test. For sporulation assays, unpaired t-tests were performed comparing the sporulation rate of two conditions. For germination and outgrowth assays, two-way ANOVA were performed. For comparison of the proportion of bright phase spores, multiple unpaired t tests were performed. For sporicidal assays, unpaired t-test were performed.

Bar plots, curves, violin plots and statistical analysis were performed with GraphPad Prism 10.0.1 (170). Figs 2A, 4B, S1D and S2A were produced with BioRender.com.

\* indicates p-value<0.05, \*\* <0.01, \*\*\* <0.001 and \*\*\*\* <0.0001.

## Supporting information

**S1 Fig. Conservation of protein sequences of Trx partners and of genetic organization of the *grd* operon.** (A) Alignment of TrxB sequences from diverse bacteria. Alignment was performed using MAFFT software [112]. The CxxC active motif is indicated by a red box and the NAD(P)H binding motifs by a green box. (B) Distance tree of TrxBs was obtained using the neighbor-joining method. FFTRs cluster in orange and NTRs in green. The atypical TrxB from *Desulfovibrio vulgaris* [114] was used to root the tree. Bootstraps are indicated on the branches. (C) Alignment of TrxA sequences from diverse bacteria. Alignment was performed using MAFFT. CxxC active domain is indicated by a red box, the classical GP site is highlighted in green and the atypical V/EP site in blue. (D) Synteny of Clostridial *grd* operons. Sequence of *grd* operons from proteolytic Clostridia were analyzed using the MicroScope platform [57].
(TIF)

**S2 Fig. Promoters of the *trx* genes and regulation of the *trxB1* gene and of genes encoding its associated ferredoxin.** (A) Promoter identification through 5'RACE using RNA extracted

from exponentially growing cells of strain 630Δ*erm*. The TSS (+1) is indicated in red. Upstream this TSS, s$^B$ boxes are represented in orange. (B-C) Expression of the *trxB1* gene and the *CD3605.1* gene encoding a ferredoxin was monitored by qRT-PCR in (B) WT strain after 24 h of growth in TY medium in anaerobiosis or at 1% O2 or in (C) WT strain and *sigB* mutant after 4.5 h of growth in TY. Experiments were performed in at least 6 biological replicates. Mean and SD are shown. One sample t-tests were used with comparison of the fold change to 1. *: p- value<0.05, ** <0.01, *** <0.001.
(TIF)

**S3 Fig. Growth curves and survival of *trx* mutants.** (A) Growth curves of the different *trx* mutants. Growth was monitored in a 96-well plate using an initial bacterial suspension at OD600nm 0.05 in TY for 24 h at 37˚C. Experiments were performed in 5 biological replicates. Mean and SD are shown. The first panel presents all curves represented in other panels as individual curves. (B, C) Survival of (B) *trxA* and (C) *trxB* mutants. A bacterial suspension at OD600nm 0.05 in TY was prepared. Total bacteria were numerated daily over 2 days by plating serial dilutions on TY Tau plates. Experiments were performed in 5 biological replicates. Mean and SEM are shown. Multiple unpaired t-tests were performed. *: p-value<0.05, ** <0.01, *** <0.001.
(TIF)

**S4 Fig. Stress tolerance of complemented strains.** (A, B) Samples were serially diluted, plated in duplicate on TY Tau plates and incubated either in anaerobiosis or in hypoxia at (A) 1% O2 or (B) 0.1% O2 for 64 h. Survival was then normalized by doing the ratio of the mutant *vs* the WT. (C, D) Samples were serially diluted and plated on TY and on (C) TY + DEA NONOate 750 μM or (D) TY + HClO 0.1% and incubated for 24 h. Mean and SD are shown. Experiments were performed in 5 biological replicates. For all assays, one-way ANOVA were performed followed by Dunett's multiple comparison tests. *: p-value<0.05, ** <0.01, *** <0.001 and **** <0.0001.
(TIF)

**S5 Fig. Conservation of the *trxB4* gene and the promoter of the *trxA2-trxB4* operon.** (A) Comparison of the *C. difficile* core genome and *trxB4* evolution. The tanglegram representing the *trxB4* tree (right) and the core-genome tree pruned to contain only genomes present in the *trxB4* tree (left). The branch lengths are measured in substitutions *per* site and are on the same scale. The only difference between the two trees is that the *trxB4* one is less resolved, which is explained by the limited phylogenetic signal in only one gene used for its reconstruction. The figure is produced with Dendroscope [111]. (B) Alignment of *trxA2* promoter region in the 630Δ*erm* and the E1 strains. Regions corresponding to the 150 bp upstream the *trxA2* start codon (ATG) and the following 100 bp from the 630Δ*erm* and the E1 strains were aligned using the MAFFT software [112]. The ATG, the TSS and the s$^A$ promoter [46] are indicated. Jalview software [113] was used for visualization.
(TIF)

**S6 Fig. Bile acids tolerance of complemented strains.** Strains were serially diluted and plated on TY and on (A) TY + DOC 0.03%, (B) TY + CHO 0.4%, (C) TY + GlyDOC 0.1%, (D) TY + Triton X-100 0.01%, (E) TY + SDS 0.003% or (F) TY + diamide 0.02% and incubated for 24 h. Survival was calculated by doing the ratio between CFUs in the last dilution with stress and CFUs in the last dilution without stress. Mean and SD are shown. Experiments were performed in 5 biological replicates. For all assays, one-way ANOVA were performed followed by Dunett's multiple comparison test. *: p-value<0.05, ** <0.01, *** <0.001 and **** <0.0001.
(TIF)

**S7 Fig. Regulation and role of the *trxA1B1* operon during sporulation. (A)** Samples were serially diluted, plated in duplicate on TY Tau plates and incubated either in anaerobiosis or in hypoxia at 1% O2 for 64 h. Survival was then normalized by doing the ratio of the mutant *vs* the WT. Mean and SD are shown. Experiment was performed in 5 biological replicates. (B) Average P*trxA1B1*-FAST^CD fluorescence intensity of different compartments from acquired images of panel 7B. Each group consists in the measure of the average P*trxA1B1*-FAST^CD of 150 cells from two independent experiments. (C) Promoter identification through 5'RACE using RNA extracted from exponentially growing cells of strain 630Δ*erm sigB::erm*. The TSS (+1) is indicated in red. Upstream this TSS, s^B box is represented in orange, s^G box in green. (D) s^G consensus, Figure from [46]. (E) Growth curves of the Δ*trxA1/B1* mutants. Growth was monitored in a 96-well plate using an initial bacterial suspension at OD600nm 0.05 in TY for 24 h at 37˚C. Experiments were performed in 5 biological replicates. Mean and SD are shown. (F) ~$10^7$ spores of WT strain and of Δ*trxA1/B1* mutant were exposed to 1% Tau to induce germination in air with 0.1% DTT. OD600nm was monitored every 5 min to evaluate germination. Experiments were performed in 5 replicates with at least 2 independent spore suspensions. Mean and SD are shown. (G) Outgrowth assay. ~$10^7$ spores of WT strain and of Δ*trxA1/B1* mutant were exposed to 1% Tau. OD600nm was monitored every 10 min for 1 h followed by every 30 minutes for 10 h. OD600nm was normalized by initial OD600nm. Experiments were performed in 5 replicates with at least 2 independent spore suspensions. Mean and SEM are shown. For O2-survival, one-way ANOVA was performed followed by Dunett's multiple comparison tests. For fluorescence intensity quantification, Kruskal-Wallis tests were performed followed by Dunn's multiple comparison test. For germination and outgrowth assay, two-way ANOVA were performed *: p-value<0.05, ** <0.01, *** <0.001 and **** <0.0001.
(TIF)

**S1 Table. List of strains used in this study.**
(PDF)

**S2 Table. List of plasmids used in this study.**
(PDF)

**S3 Table. List of oligonucleotides used in this study.**
(PDF)

**S1 Data. Raw data used to generate graph.**
(XLSX)

## Acknowledgments

We thank Johan Peltier, Auriane Monestier and Nicolas Kint for providing us the *busAA* mutant, the pMSR-ACE Δ*grdAB* plasmid and the *trxB1* mutant, respectively, Julie Le Bris for her help with the bioinformatic analysis and Joseph Mangony Mpay for his help in the optimization of sporulation assays. We are thankful to Arnaud Gautier for the development of the FAST tool and helpful discussions and to Jazmín Meza Torres for the design of an adapted FAST sequence for *C. difficile*. We thank Laurent Audry and Giulia Manina for their help with the microscopy experiments. Finally, we thank Olaya Rendueles, Basile Beaud, Simonetta Gribaldo and Bruno Dupuy for helpful discussions.

## Author Contributions

**Conceptualization:** Cyril Anjou, Isabelle Martin-Verstraete.

**Data curation:** Anna Zhukova.

**Formal analysis:** Cyril Anjou, Aurélie Lotoux, Anna Zhukova, Claire Morvan.

**Funding acquisition:** Isabelle Martin-Verstraete.

**Investigation:** Cyril Anjou, Aurélie Lotoux, Marie Royer, Léo C. Caulat, Elena Capuzzo.

**Methodology:** Cyril Anjou, Aurélie Lotoux, Anna Zhukova, Marie Royer, Léo C. Caulat, Elena Capuzzo, Claire Morvan, Isabelle Martin-Verstraete.

**Project administration:** Isabelle Martin-Verstraete.

**Supervision:** Claire Morvan, Isabelle Martin-Verstraete.

**Validation:** Cyril Anjou, Aurélie Lotoux, Anna Zhukova, Marie Royer, Léo C. Caulat, Elena Capuzzo, Claire Morvan, Isabelle Martin-Verstraete.

**Writing – original draft:** Cyril Anjou, Claire Morvan, Isabelle Martin-Verstraete.

**Writing – review & editing:** Cyril Anjou, Aurélie Lotoux, Anna Zhukova, Marie Royer, Léo C. Caulat, Elena Capuzzo, Claire Morvan, Isabelle Martin-Verstraete.

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
