## [Decision Letter · Decision Letter 0]

2 Nov 2023

Dear Dr. Martin-Verstraete,

Thank you very much for submitting your manuscript "The multiplicity of Thioredoxin systems meets the specific needs of Clostridia" for consideration at PLOS Pathogens. As with all papers reviewed by the journal, your manuscript was reviewed by members of the editorial board and by several independent reviewers. The reviewers appreciated the attention to an important topic. Based on the reviews, we are likely to accept this manuscript for publication, providing that you modify the manuscript according to the review recommendations.

The Reviewers agree that this is an important and scientifically-sound manuscript. They also agree that the manuscript is generally well written. Two of the reviewers suggest a few relatively straightforward additional experiments. Please consider adding them to a revised version of the manuscript to further strengthen the study.

Sincerely,

Bruce A. McClane

Academic Editor

PLOS Pathogens

Michael Otto

Section Editor

PLOS Pathogens

Kasturi Haldar

Editor-in-Chief

PLOS Pathogens

orcid.org/0000-0001-5065-158X

Michael Malim

Editor-in-Chief

PLOS Pathogens

orcid.org/0000-0002-7699-2064

The Reviewers agree that this is an important and scientifically-sound manuscript. They also agree that the manuscript is generally well written. Two of the reviewers suggest a few relatively straightforward additional experiments. Please consider adding them to a revised version of the manuscript to further strengthen the study.

Reviewer Comments (if any, and for reference):

Reviewer's Responses to Questions

**Part I - Summary**

Reviewer #1: The manuscript by Anjou et al. provides important new insight into the functions of thioredoxin systems in bacteria. By conducting a thorough phylogenetic analysis of the conservation and composition of the thioredoxin (Trx) thiol and protein repair pathway, the authors demonstrate that many members of the Firmicutes are distinguished by encoding multiple Trx system. In contrast, most bacteria encode only a single system thioredoxin reductase (TrxB), which may act on multiple thioredoxins (TrxA). TrxAs reduce disulfide bonds on oxidized proteins, while TrxB recycles TrxA soluble mediators.

To assess whether the multiple Trx systems encoded in C. difficile have differential functions or are simply functionally redundant, the authors systematically analyzed the function and requirement for the three strictly conserved Trx systems found in strain 630. Using genetic analyses involving single, double, and triple mutants and complementation strains, they show that the Trx1 and Trx2 systems are critical for providing resistance against low concentrations of oxygen, reactive nitrogen species, bleach, and hydrogen peroxide. This is consistent with these systems being regulated by the stress response sigma factor, SigB. They further demonstrate that these Trx systems help C. difficile to resist bile acids but not other detergents like SDS or Triton, suggesting that bile acids specifically induce unwanted disulfide bonds. Consistent with this hypothesis, the addition of the reducing agent DTT reversed the toxicity of the bile acid tested, providing critical new insight into how bile acids cause toxicity in C. difficile.

While the Trx1 and Trx2 systems help C. difficile vegetative cells resist oxygen and additional small molecule stresses, the authors’ data indicate that the Trx3 system promotes sporulation potentially by impacting the levels of the glycine reductase complex, which trxA3 is encoded within. Despite this finding, it is unclear whether the Trx3 system is found within the spore. In contrast, TrxA1 and TrxB1 have been found in the spore proteome, and the authors show that the transcription of these genes is specifically upregulated in the forespore in a SigG-dependent manner. Excitingly, they show that the TrxA1/B1 appears to affect spore resistance to bleach, suggesting that this system may be functional in dormant spores. The authors present germination data suggesting that a ∆trxA1-trxA2 deletion mutant does not germinate as fully in ambient air vs. during anaerobiosis, but this difference is quite subtle.

Finally, the authors also identify a fourth Trx system encoded in some strains of C. difficile and show that the Trx4 system can functionally complement a triple mutant strain lacking the Trx1-3 systems for O2 tolerance.

In summary, this nicely written and designed study provides novel insights into the Trx system in C. difficile, revealing a functional specialization for the multiple systems during different parts of C. difficile’s lifecycle and reveals the impact of bile acids on inducing disulfide bond stress.

Reviewer #2: This manuscript by Anjou et al. identifies and characterizes the thioredoxin systems encoded by the anaerobic pathogen C. difficile. The authors generate a series of mutants in the trxA and trxB mutants, as well as double mutants and a triple mutant, to demonstrate their thioredoxin functions and roles in tolerance to oxidative stress. The manuscript is well-written, the data are clearly presented, and the conclusions are supported by the results. The findings represent a significant advance in the understanding of oxidative stress responses by this pathogen. Minor suggestions are listed below.

Reviewer #3: The manuscript by Anjou and co-authors reports has several points in its favor. It reports on an analysis of the thioredoxin systems present across bacteria, which highlights the multiplicity of these systems in the Clostridia and then focuses on the genetically tractable human pathogen C. difficile, a strict anaerobe, to dissect their function. The findings are important in the context of infection by this organism but also of broader importance. Specifically, that at least two systems, TrxA1 and TrxA2, are important for survival in the presence of O2 and that the TrxA systems and TrxB1/B2 are important for survival in the presence of molecules that are produced during inflammation; that TrxA1/TrxB1 influence spore germination and hypochlorite resistance; that TrxA3 contributes to sporulation via a GrdAB-dependent mechanism; and the elegant observation that the Trx systems are involved in coping with the ability of bile salts to induce disulfide-bond formation.

The experimental work is carefully done. Overall, the manuscript adds important information to our knowledge on the biology of C. difficile.

**Part II – Major Issues: Key Experiments Required for Acceptance**

Reviewer #1: Since different bile acids have been shown to have differential effects on C. difficile’s physiology, e.g. deoxycholate can induce biofilm formation, it would be helpful to test an additional bile acid such as lithocholate to see if it also causes disulfide bond stress, or if the authors’ finding is more specific to deoxycholate.

Since the optical density assay measures changes in spore density across the population, the difference could be an effect across the entire population or a subset of spores that never germinate. Visualizing the germination of WT vs. ∆trxA1-trxA2 mutants using phase-contrast microscopy at a few of the timepoints (and fixing the spores at a given timepoint) would provide useful insight into whether the Trx1 system impacts germination. A more convincing phenotype may be during the outgrowth phase, where ∆trxA1-trxA2 spores would likely be killed more readily during outgrowth when exposed to 1% O2 or ambient air during germination in the presence of taurocholate and media)

Reviewer #2: n/a

Reviewer #3: It is my opinion that two experiments would add value to the manuscript, one is related to spore germination; it has several components but in reality is a single experiment. I think the extent of spore germination could be affected in a trxA1/B1 mutant because every abundant spore surface proteins are cysteine-rich. Disulfide-bond formation could thus affect spore structure and indirectly germination. If so, DTT could revert the defect (or a mutant in the gene for one of the most abundant of those proteins). Since germination is triggered by taurocholic acid, this would provide a link to the author´s finding that the thiredoxin systems are involved with coping with the formation of disulphide bonds promoted by bile acids.

The other, less important, is to look at expression of trxA3 (which is involved in sporulation) during sporulation, using the FAST reporter.

From my review letter:

1) Germination occurs without macromolecular synthesis and so it is difficult to envisage how the activity of trxA1/B1 could influence the extent of germination. Is it possible that the difference seen in the presence of oxygen is related to some structural/mechanical effect on the proteins that form the spore surface? Several of the proteins that compose the exosporium are cysteine rich and disulfide bond formation (perhaps promoted by the presence of taurocholic acid?) could perhaps induce a structural alteration of the spore surface that would interfere with the extent of germination. If so, could the addition of DTT restore the extent of spore germination to the trxA1/B1 mutant? Alternatively, a mutant in the gene coding for the most abundant of these cysteine-rich proteins, cdeM, could also eliminate differences between the wt and the trxA1/B1 mutant.

Since the germination assay was done in BHI, it should in principle measure both the initial rate of germination (decrease in the OD600) but also the outgrowth of cells from germinated spores (increase in the OD600) after a certain period of time, presumably in excess of 60 min, which is the last time point examine. Do the authors have data past this point? Are there differences in outgrowth among strains/conditions?

2) Expression of trxA3 is under the control of a sA-type promoter. But how is expression during sporulation? whole sporangium, increased in the forespore or in the mother cell. Have the authors looked at trxA3 expression using the FAST system?

**Part III – Minor Issues: Editorial and Data Presentation Modifications**

Reviewer #1: For the different mutants that they analyze, please include the “delta” ∆ symbol to indicate that it is a gene deletion. The graphs can be tricky to interpret when just referring to the gene that has been deleted.

Please provide molar concentrations for the bile acids tested

Line 101: the statement that primary bile acids allow spore germination is not technically correct because chenodeoxycholate is a primary bile acid that is a potent inhibitor of germination.

Line 105: the statement as written makes it seem like the switch to glucose fermentation leads to an increase in oxygen, but the use of the term “hypoxia” makes it seem like oxygen is scarce and that is what leads to barrier dysfunction. My understanding was that butyrate consumes oxygen, resulting in a more hypoxic environment that is better for gut barrier function?

Line 198: Since this is the first report of the use of the FAST fluorogen reporter in C. difficile to my knowledge, it would be helpful to explain more how the reporter works. The ability of the reporter to be used under strictly anaerobic conditions could be emphasized here.

Line 220: Have the authors looked under the microscope at their cells during later stages of the bile acid treatment? It is striking that they don't see the optical density decrease even though they see a 2-log decrease in CFUs.

Regarding Fig 2E, the authors mention that they previously demonstrated that SigB activity is heterogeneous, with some cells expressing the SigB-dependent transcriptional reporter and others not. However, in their graph quantifying the fluorescence of their PtrxA1-FAST reporter during exposure to 1% O2 in a WT background, they don’t see a proportion of cells that are off (i.e. around 100 fluorescence units) in contrast with a qualitative analysis of the fluorescence micrographs of the cells shown in Fig 2D. Could the authors comment on the discrepancy?

Fig 8B and 8C should use a two-way ANOVA to compare the difference between the curves rather than individual data points, since the assay involves repeated measurements on the same samples rather than discrete measurments on multiple sapmles

Well written manuscript but a few small typos or grammatical errors to be corrected:

Lines 25, 26: remove the hyphens after “thiol-” and “protein-” and “thioredoxin-“

Line 51: please modify the sentence so that it reads “The thioredoxin (Trx) system plays a crucial

Line 54: please add a hyphen so that it reads “clostridial-specific”

Line 60: please delete “an”

Line 97: please delete “an”

Line 112: the phrase “and the dispersion of newly-formed spores in the environment” seems out of place and could likely be deleted.

Line 214: please replace “but” with “except”

Line 235: please add “the” before “O2-level”

Line 242: “simple” should be “single”

Line 335: “phenotypical” could be “phenotypic”

Line 369-372: these sentences should be written in the past tense

Line 402: please consider replacing “actually” with “interestingly”

Reviewer #2: The comment in the title “meets the specific needs of Clostridia” is vague. Presumably there are several trx in these anaerobes in order to support the greater need to combat oxidative stresses that they encounter?

Ln 95: the Stickland reactions also occur outside of the Clostridia

Ln 115: There may be systems besides the Trx that lack similarity to known mechanisms and have not yet been identified.

The abbreviations NTR and FFTR should be defined.

Ln 152: manipulatable?

Ln 158: Please change “indicating” to suggests, unless there is evidence of ferredoxin cofactor.

Reviewer #3: Several minor points related to the text and the figures, taken from my review letter.

Abstract:

Line 24: on their cysteines

Lines 45-47: suggestion: “…most likely meets specific need off the Clostria in adaptation to strong stresses sporulation and Stickland pathways”

Author summary:

Lines 62-64: suggestion: the needs of cells during growth and differentiation, not only in Clostridia but perhaps in other multiple-Trx-reductase…

Introduction:

Line 78: oxidized cysteines in proteins

Line 85: Fe-S clusters

Line 101: preventing host colonization by this organism

Line 105: “increases the O2 tension in the gut” and “resulting hypoxia” is not very clear.

Line 121: the life cycle of C. difficile and their associated regulations.

Results:

Line 138: in the latter two groups

Fig 1: line 146: circles and squares

Fig 2, line 190: cells instead of bacteria

Line 193: As C. difficile faces

Line 200: induced in the presence

Line 201: fluorescent signal in the presence of O2

Line 204: In summary, the different trx loci are differentially regulated

Line 219: or multiple mutants

Line 235: the. survival in the presence of 1% O2

Line 248: the same results were obtained in. the presence of 0.1% of O2

Fig. 3, line 251: Tau could be defined here.

Fig. 3, line 254: then normalized as the ratio of

Fig. 3, line 254: Samples (not strains) withdrawn from cultures were serially diluted…

Line 281: widespread instead of spread?

Line 293: several independent gene loss events

Fig. 4, line 302: strains serially diluted?

Line 321: title, suggestion: Two Trx systems are required to cope with disulfide-bond formation induced by bile salts

Fig. 5, line 339: strains serially diluted?

Lines 369-370: why was sporulation evaluated using ethanol shock and not heat resistance tests? Also in the methods section.

Line 374: …slight. Contribution of the two other TrxAs to sporulation when TrxA3 is absent, or in the absence of TrxA3.

Line 390: since the grdAB genes are in the middle of the operon, it would be helpful here if the nature of the mutation used is specified; is it an in-frame deletion or are polar effects to be expected?

Line 403: a reference to the FAST reporter and its use could be included here that mentions its use in anaerobes, if there is one. References are given in the methods section but their inclusion here would facilitate reading. It is also important to state whether the trxA1-FAST fusion is functional. Will trxA1-FAST complement a double trxA1/trxB2 mutant, for example, for growth on plates upon exposure to 1%O2?

Line 404: what is the percentage of vegetative cells, mother cells and forespores in which accumulation of the trxA1-FAST fusion is seen? Same for the transcriptional fusion, just below.

Line 407: expression of PtrxA1B1-FAST increased in the forespore in terms of percentage or signal intensity or both? Quantification of these parameters seems important here because it would allow a correlation of the time/intensity of expression with stages in morphogenesis.

Line 410-411: why expression of the transcriptional fusion is no longer detected in phase bright forespores? Looking at the images included in Fig 7 (panels B and C), was phase contrast microscopy used or other contrast technique?

Figure 7C: cartoons on the right; sK is represented as active in the mother cell, together with sE and sB in the “mature forespore” cartoon, but not in the “complete engulfment” cartoon. Why? Also, in the bottom set of panels, sK is not represented in the sigG mutant, but there is at least some activity of sK in this mutant.

Please check scale bars in panels A and B (10 µm) and C (3 µm).

Line 444: the text mentions the consensus sequence of a promoter recognized by sG; Fig. 7D should be referred to here; panel D of Fig. 7 could also represent the accepted consensus sequence for sG-recognized promoters.

Line 450: I would prefer “mutant blocked following engulfment completion” as engulfment is a process, not a stage (also in other places throughout the text).

Line 456: unclear why the trxA1/B1 system would delay sporulation. Can the authors comment on this?

Lines 460-461: “significantly” lower reduction in OD600 suggesting a “slightly” lower efficiency of germination do not seem compatible.

Line 466: more impacted than the one of…

Line 479: calculating instead of doing.

Methods

Line 626: what is pMSR?

Line 635: the expected deletion is an in-frame deletion?

Line 641: include here a reference to the table listing the primers used.

Line 693: rhodamine.

Line 696: what is pFT47?

Line 710: after 24 h of growth in anaerobiosis

Line 739: sporulation rate was estimated by calculating the ration of spores to total cells over time.

Line 755: dilutions of cultures at midlog?

Line 744: spores were enumerated

Line 814: using the Auto Strategy function?

References:

Reference 52 is not complete.

FIGURES

Figure 2A: the trxA2/trxB4 operon, found in some strains, could also be represented.

Figure 2D: please check the scale bar. In panel E, the 600 ms does not need to be included in the title of the y axis.

Figure 6: perhaps % of sporulation is better as the title for the y axis of all panels.

Figure. 7: see comments above on phase contrast, scale bars, sG consensus and cartoons on the right of panel C. Also, the data on expression of trxA1/B1 is split between two figures (Figure 2 and Figure 7). It would be probably better to include the data in a single figure or in consecutive figures, but this may be difficult to do given the present organization of the manuscript.

Figure 8, panel B: “air” in the top panels could be replaced by aerobic conditions or equivalent.

PLOS authors have the option to publish the peer review history of their article (what does this mean?). If published, this will include your full peer review and any attached files.

Reviewer #1: No

Reviewer #2: No

Reviewer #3: No

Figure Files:

Data Requirements:

Reproducibility:

References:

---

## [Decision Letter · Decision Letter 1]

26 Jan 2024

Dear Dr. Martin-Verstraete,

We are pleased to inform you that your manuscript 'The multiplicity of Thioredoxin systems meets the specific lifestyles of Clostridia' has been provisionally accepted for publication in PLOS Pathogens.

Best regards,

Bruce A. McClane

Academic Editor

PLOS Pathogens

Michael Otto

Section Editor

PLOS Pathogens

Michael Malim

Editor-in-Chief

PLOS Pathogens

orcid.org/0000-0002-7699-2064

Reviewer Comments (if any, and for reference):

Reviewer's Responses to Questions

**Part I - Summary**

Reviewer #1: The authors have done an excellent job responding to the Reviewer's feedback. This is a very exciting manuscript.

Reviewer #2: The authors responded to all of the major criticisms of the first review and made responsive changes to the figures and text. I have no further concerns.

**Part II – Major Issues: Key Experiments Required for Acceptance**

Reviewer #1: (No Response)

Reviewer #2: (No Response)

**Part III – Minor Issues: Editorial and Data Presentation Modifications**

Reviewer #1: (No Response)

Reviewer #2: (No Response)

PLOS authors have the option to publish the peer review history of their article (what does this mean?). If published, this will include your full peer review and any attached files.

Reviewer #1: No

Reviewer #2: No

---

## [Editor Report · Acceptance letter]

5 Feb 2024

Dear Dr. Martin-Verstraete,

We are delighted to inform you that your manuscript, "The multiplicity of Thioredoxin systems meets the specific lifestyles of Clostridia," has been formally accepted for publication in PLOS Pathogens.

Best regards,

Michael Malim

Editor-in-Chief

PLOS Pathogens

orcid.org/0000-0002-7699-2064